# Tracking the hologenome dynamics in aquatic invertebrates by the holo-2bRAD approach
Cen Ma [1,2,3,7], Chang Xu[4,7], Tianqi Zhang[1,7], Qianqian Mu[4], Jia Lv[1,2], Qiang Xing[2,3], Zhihui Yang[4], Zhenyuan Xu[4], Yalin Guan[1], Chengqin Chen[1], Kuo Ni[1], Xiaoting Dai [5,6], Wei Ding[1], Jingjie Hu[1,4], Zhenmin Bao[1,3,4], Shi Wang [1,2,4] & Pingping Liu [1,2,4] ✉

The "hologenome" concept is an increasingly popular way of thinking about microbiome-host for marine organisms. However, it is challenging to track hologenome dynamics because of the large amount of material, with tracking itself usually resulting in damage or death of the research object. Here we show the simple and efficient holo-2bRAD approach for the tracking of hologenome dynamics in marine invertebrates (i.e., scallop and shrimp) from one holo-2bRAD library. The stable performance of our approach was shown with high genotyping accuracy of 99.91% and a high correlation of $r > 0.99$ for the species-level profiling of microorganisms. To explore the host-microbe association underlying mass mortality events of bivalve larvae, core microbial species changed with the stages were found, and two potentially associated host SNPs were identified. Overall, our research provides a powerful tool with various advantages (e.g., cost-effective, simple, and applicable for challenging samples) in genetic, ecological, and evolutionary studies.

No animals or plants in the wild have ever been free of microorganisms, and they all live with diverse microbiota in and around them. Microorganisms are increasingly recognized as an important component of animals and plants in virtually every environment, which affect the core functions of the host (e.g., development, immunity, nutrition, and reproduction)[1–4]. Compared with the terrestrial environment, the marine environment is more complex. Marine organisms live under harsh conditions with high salinity and pressure, limited food supply, and a lack of oxygen[5], and are mostly or completely dependent on microbes[6]. Thus, the interactions between marine organisms and microbes are of pivotal importance for understanding marine organismal adaptation and evolution. Some marine organisms, such as bivalves, are ecological keystone species that fulfill essential roles in coastal ecosystems. Their benthic filter-feeding lifestyle makes them useful sentinels to detect pollution[7]. The interactions of marine organisms and microbes can also reflect the ecological characteristics of ecosystems, playing a significant role in water quality regulation and ecological stability and influencing the growth and health of marine organisms[8,9].

An increasingly popular way of thinking about microbiome-host ecology and evolution is expressed by the concept of the hologenome. Under this concept, the holobiont (the host and its microbiota)[10,11] with its hologenome (the sum of the genetic information of the holobiont), acting in consortium, is considered to be the biological unit of natural selection[12–14]. There is now considerable evidence supporting this concept[15–17], which has broadened our perspectives for studying host-microbe coevolution. Facilitated by the advances in high-throughput sequencing technologies, studies of the hologenome have progressively increased in number, extending to a diverse range of organisms through the use of different samples (e.g., intestinal contents in chicken[18] and salmonid[19], feces and anal swab samples in vampire bat[20] and digestive gland in clams[21]). However, few hologenome studies have been performed in marine organisms, and these studies mainly focused on static hologenome data (e.g., at a specific point in time). This is a limitation given that the hologenome is dynamic, with the microbiome changing rapidly in terms of relative numbers and composition with developmental stage, physiological conditions, starvation, dietary factors, and environmental factors[22].

[1]Fang Zongxi Center for Marine Evo-Devo & MOE Key Laboratory of Marine Genetics and Breeding, College of Marine Life Sciences, Ocean University of China, Qingdao, China. [2]Laboratory for Marine Biology and Biotechnology, Laoshan Laboratory, Qingdao, China. [3]Laboratory for Marine Fisheries Science and Food Production Processes, Laoshan Laboratory, Qingdao, China. [4]Key Laboratory of Tropical Aquatic Germplasm of Hainan Province, Sanya Oceanographic Institution, Ocean University of China, Sanya, China. [5]Department of Molecular & Integrative Physiology, University of Michigan, Ann Arbor, MI, USA. [6]Institute of Gerontology, Geriatrics Center, University of Michigan, Ann Arbor, MI, USA. [7]These authors contributed equally: Cen Ma, Chang Xu, Tianqi Zhang. ✉e-mail: liupingping@ouc.edu.cn

Typically, microbial communities associated with marine organisms have been taxonomically characterized and analyzed by massive amplicon sequencing of 16 S rRNA[23–25]. While microbiota is often characterized by 16 S sequencing of bacteria, the genetic diversity of the host, fungi, and viruses is rarely investigated. Therefore, other techniques to obtain genetic information of the host need to be adopted in combination with 16 S sequencing for successful hologenomic analysis[26]. A more comprehensive approach than 16 S amplicon sequencing is shotgun sequencing, which offers the advantage of targeting the whole microbial gene pool within a sample with the possibility of running functional analysis as well as assembling genomes[27]. To date, methods for analyzing various biological information have been established to realize hologenomic analysis (e.g., simultaneous analysis of symbionts and their hosts) from one shotgun sequencing library[28,29]. However, these approaches require deep coverage of the sequencing libraries, which is associated with a high cost, and the bioinformatic processing of shotgun sequencing data is not yet standardized[7]. Moreover, destructive sampling because of the demand for a large amount of materials is also a major obstacle to dynamic hologenomic analysis.

2bRAD was described as a streamlined restriction site-associated DNA (RAD) genotyping method based on sequencing the uniform fragments produced by type IIB restriction endonucleases. The approach has originally been developed for genome-wide genotyping at minimal labor and cost[30,31]. Recently, it has demonstrated its great potential in cost-effectively producing accurate, species-resolution, landscape-like taxonomic profiles (named 2bRAD-M) for challenging microbiome samples (e.g., low-biomass, high-host-contaminated, and degraded samples)[32]. In this paper, we propose a simple and efficient approach based on 2bRAD for dynamic hologenomic analysis (named holo-2bRAD), in which hologenome data can be obtained from one 2bRAD sequencing library. We demonstrated the power of our approach in scallop (belonging to the largest marine phylum Mollusca with >100,000 extant species) and shrimp (belonging to the largest phylum Arthropoda in the animal kingdom). We evaluated and optimized the use of two kinds of noninvasive materials (gill swabs and feces), and applied a sampling method for the detection of pathogenic bacteria associated with the mortality of scallop larvae. The gill is a vital respiratory and filter-feeding organ of aquatic organisms, in which pathogens in seawater can be filter-fed and enriched[33]. Meanwhile, feces, containing the DNA of both the host and microorganisms from the intestine, can provide useful information about gastrointestinal health[34] and was also demonstrated to be an extremely valuable source of samples in hologenomic analysis[35]. Mass mortalities of bivalve larvae, referred to a sudden loss of more than 30% of the bivalve stock[36], usually occurred during the transition from the D-shaped to umbo larvae and often pose a threat to the marine fishery industry and ecosystems[37]. Certain bacteria have been reported as the causative agents of such events[38]. Thus, we tracked the hologenome dynamics of scallop larvae with different vitality at three developmental stages by using the holo-2bRAD approach. Our study provides researchers with a simple and cost-effective approach for tracking hologenome dynamics, which would be a powerful tool for efficient hologenomic analysis.

## Results and discussion
### The principle and workflow of the holo-2bRAD approach
The feasibility of the holo-2bRAD approach, which realizes the tracking of hologenome dynamics through one 2bRAD library, mainly depends on the following: (i) noninvasive sampling of key tissues to ensure the good survival status of research subjects for the continuous tracking of hologenome dynamics, (ii) co-extraction of mixed DNA containing genetic information on the host and the microbiome, a prerequisite for hologenomic analysis through one holo-2bRAD library, and (iii) a hologenome database composed of microbial taxa-specific and host-specific tags derived from microbial genomes and host genomes, which enables reliable classification of sequencing data even when mixed in a sequencing library.

The workflow of our approach can be summarized into two parts: an experimental part and a computational processing part. Specifically, the

experimental workflow started with noninvasive sampling, selecting appropriate sampling methods (e.g., wiping gill with cotton swabs or collecting feces using a pipette) to obtain materials, and then using corresponding DNA extraction methods to extract mixed DNA. Finally, a holo-2bRAD library was prepared in accordance with the instructions of the library construction method[30,31], which was then subjected to next-generation sequencing (Fig. 1).

In the computational processing workflow, the hologenome database ("holo-DB"), composed of microbial taxa- and host-specific 2bRAD tags, first needed to be established, the detailed construction method of which is described in the Methods section. Then, after preprocessing of the raw sequencing reads, the high-quality (HQ) sequencing reads were aligned to the holo-DB to identify reliable reads used for genotyping of the host and profiling of the microbiota. Finally, the sequencing 2bRAD tags derived from the host were genotyped using the RADtyping program[39] to obtain the genome-wide genotypes of the host, while tags from microorganisms were used for microbial composition and relative quantitative analysis using the 2bRAD-M pipeline[32].

### Selection of the sampling material for noninvasive sampling
There is relatively less hologenome studies in marine invertebrates, and most of studies focus on the static hologenome data[40], while dynamic hologenome data can help us to better understanding the dynamic interaction between host and microbiome to contribute to the establishment of microbial ecological strategies, and to achieve a comprehensive understanding of host physiological states, including health[6]. To obtain dynamic data, noninvasive sampling is required. Multiple tissues at the whole-body level were evaluated using gram staining to select the appropriate tissues on which to conduct hologenomic analysis in a noninvasive and efficient manner. After gram staining, although there were no swab smears up to standard with more impurities on the swab after tissue wiping according to these defined four in-house quality criteria for the assessment of stained slides[41], the results still provide us with useful information that these two tissues were rich in microorganism, suitable for hologenome analysis. There were fewer bacteria and less background noise observed after wiping the foot of shrimp and mantle of scallop (Supplementary Fig. 1). Compared with the foot swabs of shrimp, a mass of bacteria were present in the gill and feces of shrimp, although there were also more impurities. In scallop, compared with the levels in mantle, more bacteria and impurities were observed in muscle and gonad, while bacteria in gill and feces were the most abundant among several tissues. The results in both scallops and shrimps revealed that gill and feces were the most enriched in bacteria, which is consistent with previous reports[17,42,43]. There are advantages and own particularities of both materials in hologenome analysis. Gill is an important respiratory and filter-feeding organ that can provide information about organisms that enter into the host[44,45], while feces can represent the health status of the intestine and inform us of what is being excreted[46]. All of the obtained results indicate that gill and feces are the most optimal tissues or materials for noninvasive and efficient hologenomic analysis.

After determining the tissue or material for sampling, a subsequent issue is how sampling is to be performed. Feces can be directly collected after natural excretion, while how to quickly and non-destructively obtain sufficient gill materials remains a challenge. In previous studies, we proposed a method of directly cutting gill filaments[47,48], but this can damage scallops or shrimp to some extent. In this study, we instead used a noninvasive wiping method and evaluated the use of different wiping materials (i.e., filter paper and cotton swab) to obtain more tissue materials. The direct lysis (DL) method introduced in a previous study[47] allowed superior yields of high-quality genomic DNA to be obtained from small amounts of tissue, and it has been proven to be a cost-effective technique with minimal sample processing and hands-on time. Here, we optimized this approach and integrated it with a wiping sampling method to perform DNA extraction to evaluate which sampling material to choose. The yield and quality of DNA from gill of Yesso scallop (*Patinopecten yessoensis*) were assessed using two types of wiping materials (i.e., cotton swab and filter paper). Genomic DNA

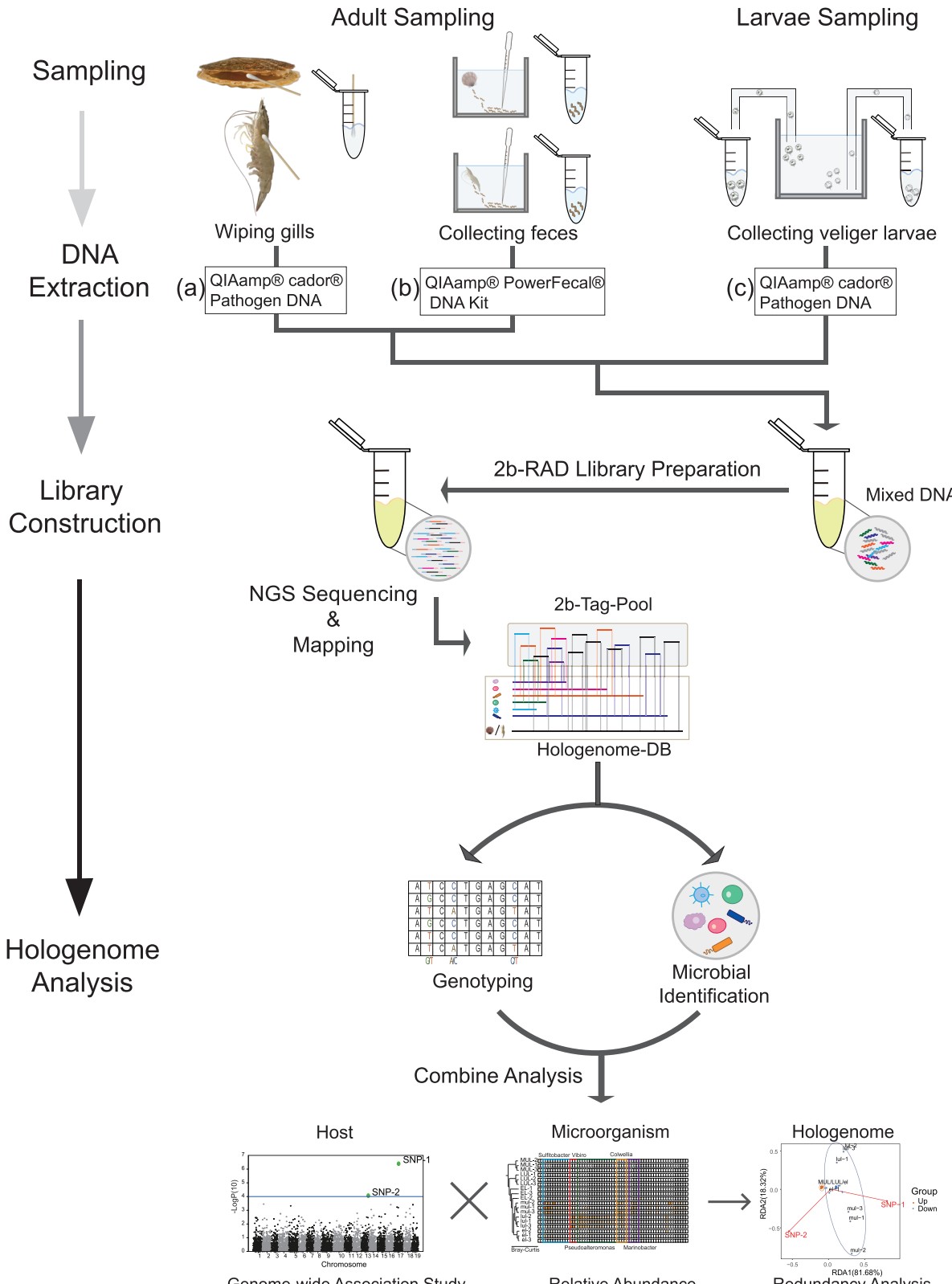

**Fig. 1 | The overview of our holo-2bRAD approach for hologenome analysis in scallop and shrimp.** The workflow involved adult sampling (including wiping gill by swabs (**a**), collecting feces (**b**) and larvae sampling (**c**), mixed-DNA co-extraction, library construction, hologenome database preparation and data analysis.

was successfully extracted from all samples with a high-molecular-weight band, as revealed by the results of agarose gel electrophoresis (Supplementary Fig. 2a). Upon comparison between the two sampling materials, samples wiped with filter paper showed less intense DNA bands. The yield of purified genomic DNA samples estimated with NanoDrop® ND-1000

(NanoDrop) was revealed to be higher (1.45–2.97 μg) upon wiping with cotton swabs, followed by that with filter paper (0.31–0.50 μg) (Supplementary Fig. 2b). It was suggested that cotton swabs performed better, producing a considerable amount of DNA with high integrity. We therefore chose cotton swabs as the sampling material.

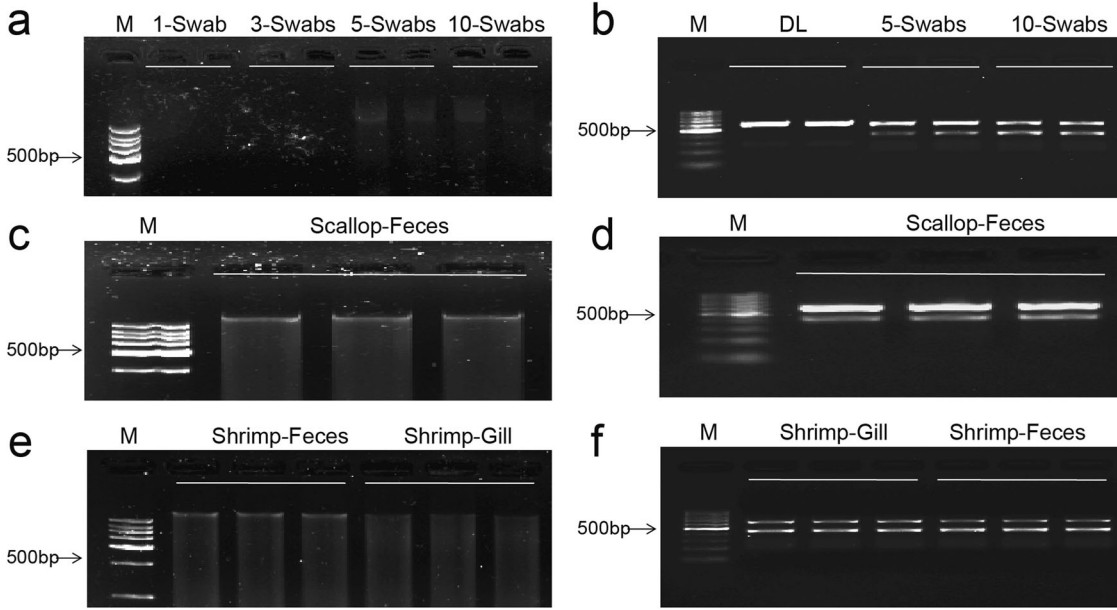

**Fig. 2 | Agarose gel electrophoresis analysis of DNA samples.** The samples were prepared by different numbers of gill swabs (**a**), feces of scallop (**c**) and shrimp (**e**), and PCR amplification products of 16 s rRNA V4-V5 region of DL, 5- and 10 swabs of gill (**b**), feces of scallop (**d**) and shrimp (**f**). 1-/3-/5-/10 swabs represents the number of cotton swabs used to wipe gill.

## Extraction of mixed DNA from different sampling materials

Co-extraction of mixed DNA is an important part of performing hologenomic analysis in a single library. Several DNA extraction methods were proposed in our study to realize the co-extraction of mixed DNA. The biomass obtained by wiping gill of scallop or shrimp is very small, especially regarding the biomass of microorganisms compared with that of the scallop or shrimp itself. Thus, we evaluated the effect of extraction with different numbers of swabs (e.g., 1, 3, 5, and 10 swabs) in scallop. The QIAamp® cador®MA Pathogen DNA process was adopted to achieve the co-extraction of mixed DNA, which could digest microbial cells to a greater extent, thereby improving the efficiency of microbial DNA extraction. The DNA bands with 1- and 3 swabs were not visible to the naked eye on agarose gels due to their extremely low concentration (e.g., 4.2 and 12.2 ng/μL as detected by a NanoDrop spectrophotometer), while the DNA samples with 5 and 10 swabs showed clear bands with high integrity, with concentrations of 35.43 and 40.45 ng/μL, respectively (Fig. 2a and Supplementary Fig. 2a). These results prompted us to directly use five swabs to wipe shrimp gills, and the DNA samples showed clear bands with concentrations of 30.25–34.63 ng/μL (Fig. 2e). Meanwhile, DNA of fecal samples of scallop and shrimp was extracted using QIAamp® PowerFecal® Pro DNA Kit, and the concentration ranged from 50.32 to 54.43 ng/μL in scallop and 47.89 to 55.78 ng/μL in shrimp. The DNA molecular band of feces can be clearly viewed on the agarose gel despite the presence of some fragmentation (Fig. 2c, e).

To check whether both host and microbial DNA were present, we conducted pre-blast of the universal primers of the 16 S rRNA region in scallop and shrimp, finding that a scallop genome sequence with a length of approximately 600 bp and a shrimp genome sequence with a length of ~590 bp can also be amplified by the V45 primer pair of 515 F and 907 R. These amplified sequences in scallops and shrimps were 18 S rRNA sequences, and similar results have also been reported in cnidarians[49]. If both host and microbial DNA exist, there will be two bands after amplification with this pair of primers: one at 400 bp for microorganisms and one at 600 bp for scallop or 590 bp for shrimp. In the evaluation of DNA samples using the DL method, the agarose gel electrophoresis analysis revealed only one band (~600 bp), indicating that there was almost no microbial DNA (Fig. 2b). DNA samples of 5 and 10 swabs and feces in scallop were amplified by using the primer pair of 515 F and 907 R, and the agarose gel electrophoresis analysis revealed two bands (e.g., 400 and 600 bp), indicating the presence of both host and microbial DNA (Fig. 2b, d). DNA samples of 5 swabs and feces in shrimp were also amplified by the universal primers 515 F and 907 R, and two DNA bands were observed after the amplification (Fig. 2f), demonstrating the successful co-extraction of mixed DNA.

## Construction of hologenome database (holo-DB) in scallop and shrimp

The hologenome database of the studied species, which contains the reliable enzyme-digested sequence tags specific to high-resolution microbial taxa and host, should be constructed in advance of data analysis according to the methods we described before. First, a total of 173,165 microbial genomes, including 25,260 bacterial, 614 archaea, and 289 fungal complete genomes, from NCBI RefSeq (Oct 2019) were downloaded, and digital restriction digestion of all of these genomes by BsaXI was performed, producing 1231.48 tags per genome. Among these tags, desired holo-2bRAD tags, namely, microbial taxa-specific DNA markers that only occur once per genome, were retained. Overall, among the different taxonomic levels, the higher the taxonomic level, the more available tags were retained (Fig. 3a). At the kingdom level, very few holo-2bRAD tags are shared between bacteria, fungi, and archaea; thus, almost all holo-2bRAD tags are kingdom-specific. The phylum-, family-, and genus-specific holo-2bRAD tags account for 99.70%, 99.50%, and 99.10% of all theoretical holo-2bRAD tags generated by a given microbial genome, respectively. Holo-2bRAD markers unique to 26,163 microorganisms were also explored, revealing that, among all 212,110,011 restriction fragments generated by BsaXI, ~98.02% are single copies within a given microbial genome. Then, digital restriction digestion of host genomes (e.g., scallop and shrimp) was performed. In total, 366,747 and 773,149 non-redundant tags were obtained in scallop and shrimp, and 169,582 and 653,495 unique tags were obtained after the de-redundancy, respectively (Fig. 3a). Finally, the microbial taxa-specific tags and host-unique tags were used again to remove redundancy to construct hologenome DB (holo-DB). In the holo-DB of scallop, only 111 tags of scallop, accounting for 0.07% of all non-redundant tags, were removed; and approximately 8218 microbial genomes, accounting for 4.75% of all microbial genomes, were removed. The removed microbial genomes had almost no impact at the genus to phylum level (e.g., 0.01% to 0.2%), only

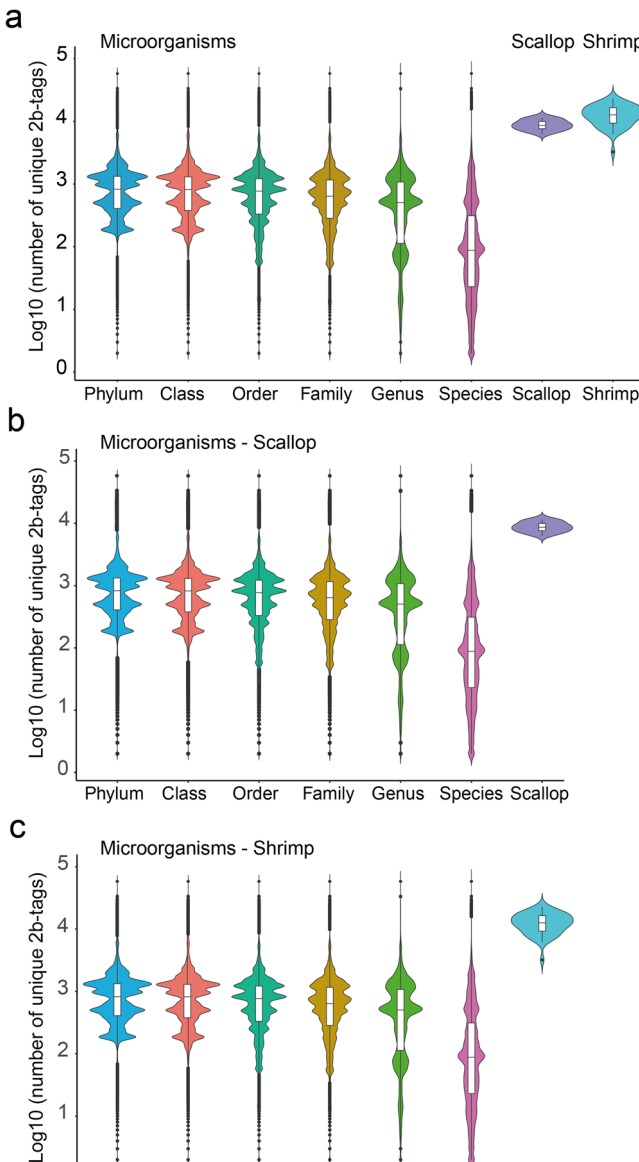

**Fig. 3 | Distribution of theoretically existent specific-2bRAD tags at various taxonomy levels of microbiota and unique-RAD tags. a** Host (e.g., scallop and shrimp), **b** specific 2bRAD tags in both microbiota and scallop after secondary re-redundancy, **c** specific 2bRAD tags in both microbiota and shrimp after secondary re-redundancy.

affecting 3.71% of species (Fig. 3b. Similar results were observed in holo-DB of shrimp with 243 tags of shrimp removed, accounting for 0.04% of all non-redundant tags, and 8329 microbial genomes dislodged, accounting for 4.81% of all microbial genomes, which affect 3.86% of species (Fig. 3c). Remarkably, none of the species were removed during the database construction, indicating that the removal of these microbial genomes has no effect at the species level. Taking the findings together, very few tags are shared between scallop/shrimp and microorganisms, which have almost no influence on the genotyping of scallop/shrimp and profiling of the microbiome. This in turn indicates the feasibility of conducting hologenomic analysis through one holo-2bRAD library.

## Performance of hologenomic analysis in scallop and shrimp

To evaluate the performance of our hologenomic approach, DNA samples of gill swabs and feces were used to construct holo-2bRAD libraries. In scallop,

the DNA sample of muscle of the same individual was also used to construct a holo-2bRAD library as a control. Sequencing these libraries produced a similar number of raw reads in the same species (Table 1). After quality filtering, more than 97% of the sequenced reads were retained as high-quality reads, which was comparable to the rate in the control group (98.41–98.76%). The high-quality reads containing the BsaXI recognition site were aligned to the hologenomic dataset of scallop or shrimp, with a mapping rate of 94.74–97.70% to scallop/shrimp and 2.30–5.26% to microbiomes in the gill-swab samples, and 88.88-–90.67% to scallop and 9.33–10.13% to micro-biomes in the fecal samples. The "kitome" experiment was conducted as a negative control to assess the potential impact of contamination in different types and batches of kits and sampling tools in the experiment. However, the concentration of "kitome" DNA was extremely low with 0.01–0.02 ng/μL, and the construction of libraries was unsuccessful for both the holo-2bRAD and 16 S amplicon sequencing. Thus, we concluded that the potential impact of "kitome" contamination in our samples can be neglected.

The genotyping performance of the host in several key aspects (e.g., genome coverage, calling rate, genotyping agreement of technical replicates, and genotyping accuracy) was first evaluated. Coverage of detected sites across the genome was examined, revealing uniform coverage in all samples (Fig. 4a). Sequencing depth of each tag across technical replicates was highly consistent, with a high Pearson correlation coefficient (Pearson's $r > 0.94$, $P < 0.05$) (Fig. 4b). The large majority of detected sites (90.03–94.87%) of 5-swab and fecal samples were shared with control samples in scallop (Supplementary Tables 1 and 2). High genotyping agreement between replicates was observed with 99.86% and 99.90% in gill and feces samples, respectively (Supplementary Table 3). Genotyping accuracy between the 5-swab and control samples, and high genotyping accuracy was also observed between fecal and control samples (99.88% and 99.87%) (Table 2). The obtained genotyping accuracy was comparable to that reported in previous 2bRAD-based studies[50,51], suggesting that it can provide satisfactory genotyping performance. Genotyping results of shrimp revealed that a total of 10,665,018 loci were genotyped in all samples (gill: 9,428,912; feces: 9,627,653), and the phylogenetic tree was constructed using all SNPs of the six samples, including gill swabs and feces. The different tissues/materials of the same individual clustered together as expected, with Sample 3 defined as a major branch separate from the other individuals.

Then, the reproducibility was analyzed to probe the ability of our approach to consistently profile the microbial community in scallop. High similarity (~95%) in the number of detected species was observed between the two technical replicates in both gill and fecal samples (Table 3 and Fig. 4c). Upon analyzing the relative abundance, there was no marked difference in the abundance of detected species between two replicates (Fig. 4e), as shown by the correlation analysis results (Pearson's $r = 0.9996$ for the gill samples and Pearson's $r = 0.9999$ for the fecal samples, $P < 0.05$) (Fig. 4d). The high reproducibility suggested the stable performance of the metagenomic analysis. In shrimp, a total of 509, 822, and 427 species were detected in gill samples, including 15 phyla of bacteria and 2 phyla of fungi. More species were detected in feces, numbering 535, 328, and 1098 in three individuals, including 19 phyla of bacteria and 1 phylum of fungi. Also, 16 S analysis was used to validate the accuracy of holo-2bRAD for microbial classification in both scallop and shrimp. The high consistency of the genus detected (scallop feces: 94%; shrimp feces: 88%; shrimp gill: 92%) and relative abundance (Pearson's $r$: scallop feces = 0.9367; shrimp feces: 0.9265; shrimp gill: 0.9298 ($P < 0.05$); L2- similarity: scallop feces = 0.9154; shrimp feces = 0.9063; shrimp gill: 0.9143) indicated that the holo-2bRAD was accurate for microbial identification (Supplementary Fig. 3).

In accordance with the SNP phylogenetic tree (Fig. 5a), the hierarchical clustering of identified microbial species in shrimp revealed that the same type of material clustered together, while in the same tissue/material, Sample 1 and Sample 2 were more similar (Fig. 5b). All of these results demonstrated that there are differences in microbial composition among individuals despite them having the same age and growing environment. Individuals with more similar host genotypes tend to have more similar microbial composition[52,53], suggesting that the host genetic variation may be associated

**Table 1 | Sequencing statistics of holo-2bRAD libraries of gill/feces and control samples in scallop and shrimp**

| | Scallop gill | | | Scallop feces | | |
|---|---|---|---|---|---|---|
| | **Muscle** | **Rep 1** | **Rep 2** | **Muscle** | **Rep 1** | **Rep 2** |
| Raw reads | 11,356,587 | 10,838,115 | 10,827,205 | 10,096,799 | 10,496,678 | 10,533,265 |
| High-quality reads | 11,143,083 | 10,630,023 | 10,556,324 | 9,971,598 | 10,203,820 | 10,142,807 |
| High-quality reads rate | 98.41% | 97.15% | 97.05% | 98.76% | 97.21% | 97.05% |
| HQ tags with the restriction site | 9,211,275 | 9,050,087 | 8,885,492 | 9,043,375 | 8,311,095 | 8,302,067 |
| Mapping rate (microorganism) (%) | 0.01% | 2.32% | 2.30% | 0.01% | 10.06% | 10.13% |
| Mapping rate (scallop) (%) | 99.99% | 97.68% | 97.70% | 99.99% | 89.94% | 89.87% |
| | Shrimp gill | | | Shrimp feces | | |
| | **Sample 1** | **Sample 2** | **Sample 3** | **Sample 1** | **Sample 2** | **Sample 3** |
| Raw reads | 76,359,818 | 71,720,596 | 65,356,084 | 76,729,108 | 75,291,073 | 62,540,880 |
| High-quality reads | 75,747,320 | 71,352,591 | 64,855,236 | 76,268,255 | 74,939,215 | 62,309,791 |
| High-quality reads rate | 99.20% | 99.49% | 99.23% | 99.40% | 99.53% | 99.63% |
| HQ tags with the restriction site | 65,382,266 | 68,276,165 | 58,286,956 | 57,580,070 | 62,874,355 | 49,927,225 |
| Mapping rate (microorganism) (%) | 5.26% | 5.18% | 4.69% | 9.67% | 11.12% | 9.33% |
| Mapping rate (scallop) (%) | 94.74% | 94.82% | 95.31% | 90.33% | 88.88% | 90.67% |

with the microbiota profile. Comparison with the microbial composition detected in gills and feces was also conducted in shrimp, with the results showing that the evenness of microorganisms detected in gills was significantly higher than that in feces ($t$ test, $P < 0.05$). The Simpson index and Shannon index were also revealed by α-diversity analysis (Fig. 5c, d).

Microbiome taxonomic profiling typically adopts two strategies, including amplicons of phylogenetic "marker genes" (e.g., 16 S rRNA for bacteria and archaea, and 18 S rRNA or internal transcribed spacer (ITS) for fungi) and the whole genomes (whole metagenome shotgun; WMS). In fact, a detailed comparison between 2bRAD approach, 16 S amplicon sequencing and WMS sequencing has been conducted in our previous study[32]. Only 1–5% of the sequencing data of shotgun approach were required by 2bRAD approach to produce a taxonomic profile of equivalent accuracy, resulting in a cost reduction of 20-100 folds compared to WMS. Compared with the resolution of 16 S amplicon sequencing at the genus level and its ability to only identify bacteria, 2bRAD approach can accurately generate species-level bacterial, archaeal, and fungal profiles. Even for these low-biomass, highly degraded, or heavily contaminated samples, 2bRAD approach exhibited excellent performance. All these advantages have been verified in many other studies, such as the low-biomass samples[54], high degradation samples[55,56], and high-host contamination samples[57,58]. Except that, in terms of algorithm, 2bRAD approach also make the improvement with combining merits from both DNA-to-Marker and DNA-to-DNA methods to perform species identification and abundance estimation, making it exhibit excellent performance in eliminating false positives[59]. Holo-2bRAD retains all the advantages of 2bRAD-M in microbial analysis, while also allows the precise host SNP genotyping, which was not involved in 2bRAD-M. Holo-2bRAD relied on the existing microbial genome database, making it difficult to conduct if the microbial community is unknown. This can be solved by using WMS to de novo assemble the genomes or MAGs of the sample.

**Tracking the hologenomic dynamics in scallop larvae with different vitality**

Mass mortalities of scallop larvae occurred in a hatchery (MM) in Rongcheng, Shandong Province, China, and scallop larvae of three stages in the upper/lower layers in MM were sampled, including three developmental periods (middle-umbo larvae [MUL/mul], late-umbo larvae [LUL/lul], and eyespot larvae [EL/el]; uppercase/lowercase letters represent larvae living in the upper/lower layers) (Supplementary Fig. 4). For these challenging larval DNA samples with either low biomass or degradation (Supplementary Fig. 2), holo-2bRAD libraries were successfully constructed. After mapping

to the hologenome database, a significant difference in the proportion of microbial tags between the upper and lower layers in MM was found ($P < 0.05$) using Mann–Whitney $U$ test (Supplementary Fig. 5a). Compared with the findings in the upper layer, the larvae that were moribund or dead in the lower layer accounted for a higher proportion of microorganisms, among which the highest proportion of microorganisms was detected in the mul stage, followed by lul, and the proportion of microorganisms at the el stage significantly declined (Supplementary Fig. 4a). Using RADtyping software to obtain genotypes, a total of 48,820 SNPs were detected in the upper larvae and 48,174 SNPs in the lower larvae, among which 43,234 SNPs were shared between the upper and lower larvae. A phylogenetic tree was constructed using SNPs of all 18 samples in MM. It was found that the upper/lower larvae were still clustered into one branch according to the breeding tank from which they were sampled. No separation between individuals based on their particular viability was identified.

The microbiota in the three developmental stages of larvae in MM was also analyzed. For the larvae that were moribund or dead in the lower layer of tanks, there were more microorganisms detected at the mul stage, while the numbers of species detected in lul and el declined. In the three stages, Proteobacteria (82.86 ± 4.00%, 98.55 ± 0.85%, and 89.33 ± 2.45% in three stages) and Bacteroidetes (16.82 ± 3.96%, 1.40 ± 0.43%, and 9.14 ± 2.23% in three stages) were the dominant phyla, including Gammaproteobacteria, Alphaproteobacteria, and Flavobacteria with different ranking orders as the main classes across the three developmental stages. In addition, the upper larvae with good vitality contained relatively few microbial species, with Proteobacteria (91.43 ± 6.09%, 99.21 ± 1.36%, and 85.19 ± 2.15%), Actinobacteria (6.23 ± 4.40%), and Bacteroidetes (9.58 ± 4.70%) as the dominant phyla.

**Dynamic changes of hologenome in scallop and potential SNPs associated with larval viability**

The diversity and evenness of microbial communities in the three developmental stages of the upper and lower larvae collected from the culturing hatchery, which experienced mass mortalities of larvae (MM), were evaluated using α-diversity analysis, including Shannon and Simpson indexes. The Simpson indexes of the lower larvae in the three stages were significantly higher than those of the upper larvae (Supplementary Fig. 5b), with 0.94 ± 0.03 in the lower larvae and 0.66 ± 0.18 in the upper larvae. This indicated that there was higher diversity among the lower larvae than among those samples derived from the upper layer. Upon comparison among the three stages, MUL/mul showed the highest Simpson index, followed by LUL/lul and EL/el, irrespective of whether the lower or upper larvae were being evaluated. These differences among the three stages between the lower and

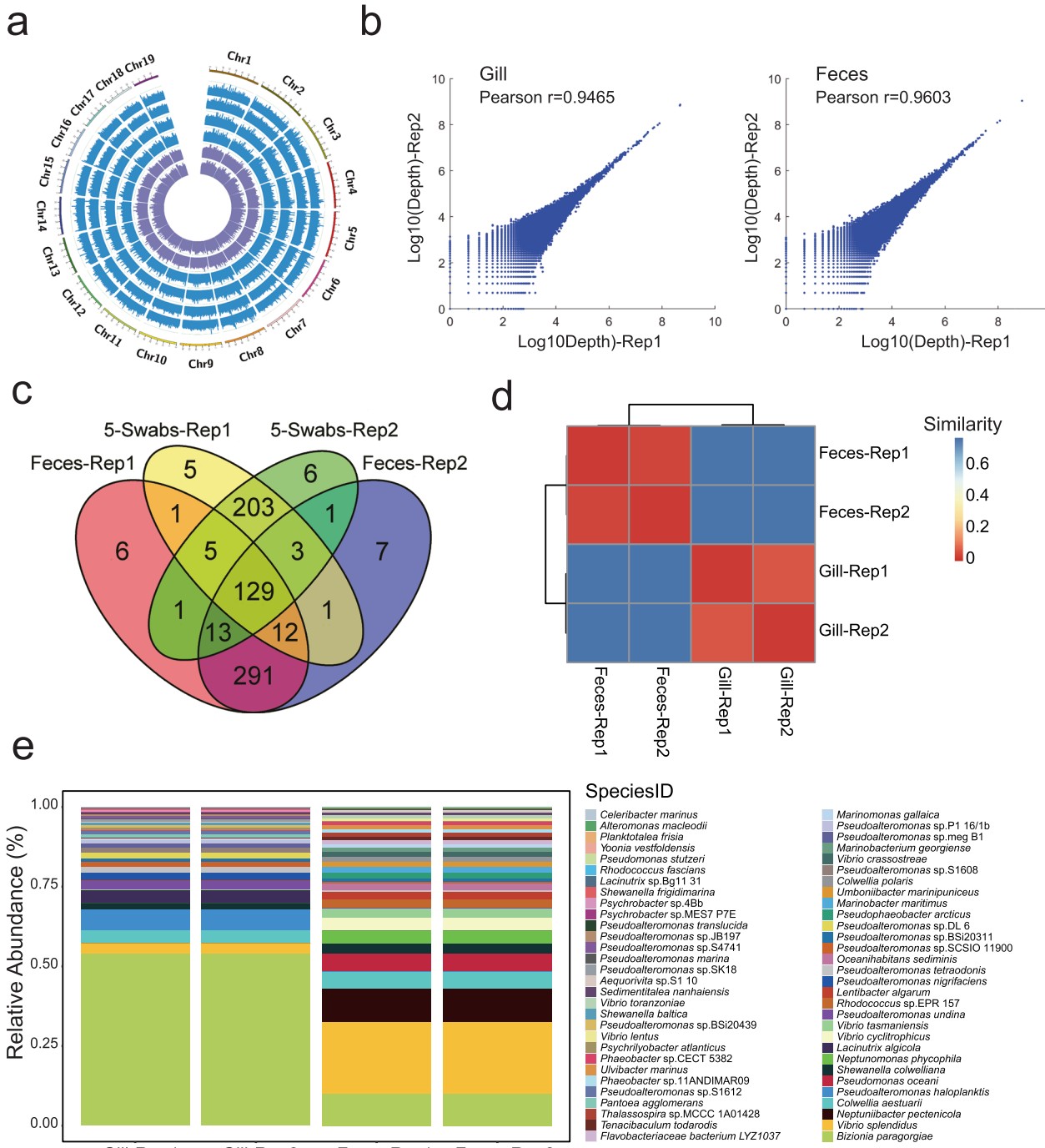

**Fig. 4 | Performance of hologenomic analysis in scallop. a** Genome representation and sequencing coverage of 2bRAD library of gill and feces of scallop. Inner purple circle 1: sequencing data generated using control-1 sample (control for fecal sample); inner purple circle 2: sequencing data generated using control-2 sample (control for gill swabs); outer blue circles: sequencing data generated using the sample of our approach (feces and gill swabs). **b** Correlation of sequencing depth of the sequencing coverage of 2bRAD tags across technical replicates in gill swabs and feces (Pearson $r = 0.9465$ and 0.9603). **c** Shared microbiological species between sample of 5 swabs and feces and their technical duplicate library of the scallop. **d** β-diversity heatmap of gill swabs and feces and their technical replicates. **e** Profiling stack column of gill swabs and feces at the species level.

upper larvae were also confirmed by the calculations of the Shannon index (Supplementary Fig. 5b). Principal component analysis (PCA) and principal coordinate analysis (PCoA) using Bray–Curtis distance matrices were performed to illustrate the dissimilarity of the bacterial community structure among the different samples. The bacterial communities of upper larvae were remarkably distinct from those of the lower larvae, with 57.16% of the total variation being explained by the two main ordinates. The results revealed that the upper larvae of the three stages were clustered together, while the lower larvae were clearly separated into three clusters according to the different developmental stages (Fig. 6a, b). A total of 18 samples were also collected in the other hatchery (NMM) that did not experience mass mortalities, and PCA and PCoA were performed using all 36 larval samples (e.g., 18 samples collected in MM and 18 in NMM). The results showed that samples collected in NMM clustered together and no segregation occurred (Fig. 6c, d). Although the upper larvae in MM and all larvae in NMM were divided into two clusters, they were significantly closer together than for the lower larvae.

**Table 2 | Genotyping accuracy between gill/feces and control sample of scallop**

| | | Homozygote | Heterozygote | All |
|---|---|---|---|---|
| Gill | Genotyped | 2,258,257 | 12,991 | 2,271,248 |
| | Same genotype | 2,256,585 | 12,014 | 2,268,599 |
| | Different, genotype | 1672 | 977 | 2649 |
| | Agreement (%) | 99.93% | 92.48% | 99.88% |
| Feces | Genotyped | 2,148,389 | 12,538 | 2,160,927 |
| | Same genotype | 2,146,476 | 11,573 | 2,158,049 |
| | Different, genotype | 1913 | 965 | 2878 |
| | Agreement (%) | 99.91% | 92.30% | 99.87% |

The species with the highest abundance at different stages were then counted. For the lower larvae, the top three abundances of species in the mul stage were *Sulfitobacter geojensis* (8.54 ± 1.22%), *S. dubiu* (3.02 ± 1.08%), and *S. indolifex* (2.41 ± 1.02%), all belonging to the same genus. And significant difference was observed in these three species accoss live stages (*S. geojensis*: $p = 0.0461$; *S. dubiu*: $P = 0.0251$; *S. dubiu*: $P = 0.0409$, Kruskal–Wallis test). In addition to the genus *Sulfitobacter*, the relative abundances of *Colwellia* and *Marinobacter* were also high in the mul stage, while the relative abundances of these three genera decreased to a low level as the larvae grew (Fig. 6e). In the lul stage, *Vibrio cyclitrophicus* became the most abundant species (relative abundance 4.66 ± 1.39%), followed by *V. splendidus* (4.01 ± 2.08%), which belong to the same genus. At the same time, high abundance of the genus *Pseudoalteromonas* was also detected, with relative abundance of 17.91 ± 4.18% in the lul stage, probably due to the antagonistic effects of *Vibrio* (average relative abundance Pearson's $r = 0.9671$, $P < 0.05$). Although the *Vibrio* genus was still the core component of the microbiome in the eyespot stage, its abundance decreased at this stage (Fig. 6e). Similarly, the genus *Sulfitobacter* is also the core member of the microbiome in the el stage, while its abundance is lower in the mul stage (Fig. 6e). Compared with the lower larvae, the abundance of detected species of the upper larvae was significantly reduced, and there was no significant difference among the three stages. The core species of the upper larvae during the three stages is *Brevundimonas* sp. SH203 (1.44 ± 0.05%, $P = 0.3651$), which has been reported as a strain capable of degrading cellouronate[60]. Some pathogens (e.g., *V. splendidus*) also dominated in MUL (0.24 ± 0.04%) and LUL stages (0.20 ± 0.03%) in the upper larvae, probably due to a status as an opportunistic bacterial pathogen of bivalves[61].

GWAS using the 49,482 informative SNPs in MM was performed to identify potential SNP markers associated with larval viability. The results showed that two SNPs exhibited a strong correlation with the viability of the larvae (Supplementary Fig. 5c). One of these was located in the intron region of the gene encoding dystrophin protein (Chr17), which has been reported to be positively correlated with the growth of scallop muscle[62]. The other was located in the intergenic region of the gene encoding mannose receptor, C-type protein (Chr13), which has been shown to be negatively correlated with innate immunity in scallop[63]. Next, the correlations (SNP-1: $R^2 = 0.5069$, $P = 0.001$; SNP-2: $R^2 = 0.4448$, $P = 0.004$) between the two SNPs and the associated microorganisms were confirmed by redundancy analysis (RDA) (Supplementary Fig. 5d).

In conclusion, we have successfully developed a simple and efficient approach for hologenome analysis of aquatic invertebrates. Our research provides a powerful tool with various advantages (e.g., cost-effective, simple, and applicable for challenging samples) in genetic, ecological, and evolutionary studies.

## Methods
### Materials and Gram staining
A total of 15 individuals of 2-year-old Yesso scallops (*Patinopecten yessoensis*) were collected from Qingdao, Shandong Province, China, and 3 individuals of 110-days-old shrimps (*Litopenaeus Vannamei*) were collected from Sanya, Hainan Province, China. All the individuals were cultured in sterile seawater which was filtered through 0.1-μm filter paper for one day to stabilize the condition, of which the individuals used for fecal sampling were cultured separately to collect feces excreted in sterile seawater. The larvae of Yesso scallops were sourced from the cultured scallop farm (Yantai, Shandong Province, China) in February of 2023. We sampled and evaluated several tissues that could successfully obtained without killing or even injuring the host. The candidate tissues in scallop were mantle, gonad, muscle, gill and feces; and in shrimp, feet, gill and feces were sampled. Wiping some of tissues (e.g., mantle, gonad, muscle and gill in scallop) need to force the shell open to a gape of ~10 mm. The swabs were gently drawn along the surface of tissues (each tissue was wiped three times using one swab), then each swab was removed and rolled onto a glass slide while avoiding clumping. In order to have an advanced knowledge of the microbial enrichment of these tissues, swab smears were stained using the Gram staining method. All slides were then Gram-stained and examined microscopically.

### Sampling methods
Two noninvasive DNA sampling methods were adopted, including wiping gill with different materials (cotton swabs and filter paper) and collecting feces, and one method to sample scallop larvae.

**Wiping sampling**. For wiping, we selected two kinds of wiping materials (cotton swabs and filter paper) to wipe gill of the scallop. A total of 12 individuals with good activity and similar size, which were divided into two groups as cotton swabs wiping gill (CG) and filter papers wiping gill (FG), were used for wiping after 1 day's cultivation in the laboratory. In order not to cause any damages to scallops, our sampling methods consisted of gently forcing the shell open to a gap of ~10 mm and then wiping gently on the surface of gill by cotton swabs or filter paper. The shrimps' gills were swabbed with cotton swabs, and in order not to cause damage to the shrimp, the shell of the head was also opened to a gap of about 10 mm. The cotton swabs and filtered paper were either used immediately or frozen in a standard commercial freezer (−80 °C) until further use. The materials used for wiping were sterilized.

**Feces sampling**. Feces sampling was also an option especially for the smaller shellfish or the species which are not so easy to wipe their tissues. Fecal samples were collected immediately using pipettes after defecation and stored at −80 °C in a 1.5 mL lyophilized tube until processing. A total of three scallops and three shrimps were used to collect feces.

**Scallop larvae sampling**. In order to explore the causes of high mortality of scallop larvae, the poor-activity larvae at the lower layer of tanks (30 m³) in two culturing hatcheries (e.g., one experienced mass mortalities of larvae, named as MM, while the other not, named as NMM) during three development periods (middle-umbo larvae (mul), late-umbo larvae (lul) and eyespot larvae (el)) were collected, and the healthy/active larvae at the upper layers of tanks were also collected for comparison (MUL, LUL and EL). Approximately 2 L seawater from the upper and lower layers were filtered through 120-μm bolting silk to collect the larvae, respectively, and the collected larvae were stored in a −80 °C refrigerator until use. Larvae from a total of six tanks were collected for study.

### DNA extraction
To meet different research purposes, different DNA extraction methods were adopted. The details were as follows:

**Direct lysis method (DL)**. Direct lysis method, a rapid and efficient DNA extraction method which bypass many tedious and time-consuming steps, were used to extract the DNA from all wiping samples. In order to co-extract the DNA of scallop and microorganism, we added appropriate

**Table 3 | Summary of the detected microbial organisms in scallop and shrimp (phylum level)**

| | | Scallop | | | | | | | | Shrimp feces | | | | | | Shrimp gill | | | | | |
|---|---|---|---|---|---|---|---|---|---|---|---|---|---|---|---|---|---|---|---|---|---|
| | DL | Gill-Rep 1 | | Gill-Rep 2 | | Feces-Rep 1 | | Feces-Rep 1 | | Shrimp-S1 | | Shrimp-S2 | | Shrimp-S3 | | Shrimp-S1 | | Shrimp-S2 | | Shrimp-S3 | |
| | No. | No. | Rate | No. | Rate | No. | Rate | No. | Rate | No. | Rate | No. | Rate | No. | Rate | No. | Rate | No. | Rate | No. | Rate |
| **Bacteria** Gammaproteobacteria | 4 | 160 | 49.38 | 160 | 48.78 | 276 | 61.88 | 274 | 61.85 | 143 | 97 | 136 | 98.5 | 148 | 91.6 | 38 | 44.9 | 75 | 20.43 | 38 | 49.87 |
| Bacteroidetes | – | 97 | 29.94 | 98 | 29.88 | 83 | 18.61 | 84 | 18.96 | 37 | 0.25 | 37 | 0.09 | 168 | 1.25 | 112 | 6.54 | 157 | 14.9 | 44 | 3.27 |
| Alphaproteobacteria | – | 40 | 12.35 | 43 | 13.11 | 63 | 14.13 | 62 | 14 | 201 | 1.23 | 75 | 0.39 | 468 | 4.66 | 233 | 26.55 | 388 | 40.26 | 276 | 33.4 |
| Actinobacteria | – | 14 | 4.32 | 15 | 4.57 | 13 | 2.92 | 14 | 3.16 | 92 | 0.37 | 21 | 0.11 | 202 | 1.26 | 50 | 3.89 | 102 | 10.31 | 24 | 1.64 |
| Firmicutes | 1 | 4 | 1.24 | 4 | 1.22 | 1 | 0.22 | 1 | 0.23 | 23 | 0.74 | 20 | 0.47 | 29 | 0.68 | 21 | 9.44 | 17 | 2.33 | 11 | 1.99 |
| Epsilon-proteobacteria | – | 2 | 0.62 | 2 | 0.61 | 5 | 1.12 | 4 | 0.9 | 23 | 0.09 | 27 | 0.26 | 6 | 0.03 | 3 | 0.05 | 4 | 0.11 | – | – |
| Cyanobacteria | – | 2 | 0.62 | 2 | 0.61 | – | – | – | – | 1 | 0.01 | – | – | 2 | 0.01 | 2 | 0.04 | 2 | 0.09 | 2 | 0.12 |
| Fusobacteria | – | 2 | 0.62 | 1 | 0.31 | 2 | 0.45 | 1 | 0.23 | – | – | 3 | 0.01 | 1 | 0.01 | – | – | – | – | – | – |
| Deinococcus-Thermus | – | 1 | 0.31 | 1 | 0.31 | – | – | – | – | – | – | – | – | 1 | 0.01 | 1 | 0.01 | 1 | 0.017 | – | – |
| Verruco-microbia | – | 1 | 0.31 | 1 | 0.31 | 1 | 0.22 | 1 | 0.23 | – | – | – | – | 4 | 0.04 | 5 | 3.13 | 7 | 1.85 | 7 | 5.36 |
| Ascomycota | – | – | – | – | – | 1 | 0.22 | 1 | 0.23 | – | – | – | – | – | – | 2 | 0.04 | – | – | – | – |
| Planctomycetes | – | – | – | – | – | – | – | – | – | 3 | 0.11 | 3 | 0.04 | 30 | 0.23 | 11 | 0.52 | 35 | 4.77 | 9 | 0.36 |
| Beta-proteobacteria | – | – | – | – | – | – | – | – | – | 4 | 0.01 | 2 | 0.01 | 17 | 0.07 | 1 | 1.79 | 8 | 0.41 | 2 | 0.17 |
| Euryarchaeota | – | – | – | – | – | – | – | – | – | 3 | 0.01 | 1 | 0.01 | 7 | 0.01 | 21 | 1.76 | 16 | 0.95 | 9 | 0.43 |
| Mucoromycota | – | – | – | – | – | – | – | – | – | – | – | – | – | – | – | – | – | – | – | – | – |
| Delta-proteobacteria | – | – | – | – | – | – | – | – | – | 2 | 0.01 | – | – | 7 | 0.01 | 1 | 0.09 | 2 | 0.07 | 1 | 0.09 |
| Oligoflexia | – | – | – | – | – | – | – | – | – | 1 | 0.02 | 1 | 0.02 | 1 | 0.01 | 1 | 0.16 | 2 | 0.21 | 3 | 0.2 |
| Rhodo-thermaeota | – | – | – | – | – | – | – | – | – | – | – | – | – | – | – | 1 | 0.01 | 2 | 0.02 | – | – |
| Chlamydiae | – | – | – | – | – | – | – | – | – | 2 | 0.01 | 2 | 0.01 | 2 | 0.01 | 1 | 0.02 | 1 | 0.1 | – | – |
| Thermotogae | – | – | – | – | – | – | – | – | – | – | – | – | – | 1 | 0.01 | – | – | – | – | – | – |
| Kiritimatiellaeota | – | – | – | – | – | – | – | – | – | – | – | – | – | 1 | 0.01 | – | – | 1 | 0.02 | – | – |
| Chlorobi | – | – | – | – | – | – | – | – | – | – | – | – | – | 1 | 0.011 | – | – | 1 | 0.02 | – | – |
| Balneolaeota | – | – | – | – | – | – | – | – | – | – | – | – | – | 1 | 0.01 | – | – | – | – | – | – |
| **Fungi** Basidiomycota | 1 | 1 | 0.31 | 1 | 0.31 | – | – | – | – | – | – | – | – | 1 | 0.01 | 3 | 0.12 | 1 | 0.01 | 1 | 0.02 |
| Ascomycota | – | – | – | – | – | 1 | 0.22 | 1 | 0.23 | – | – | – | – | – | – | 2 | 0.04 | – | – | – | – |
| **Total** | 6 | 324 | | 328 | | 446 | | 443 | | 535 | | 328 | | 1098 | | 509 | | 822 | | 427 | |

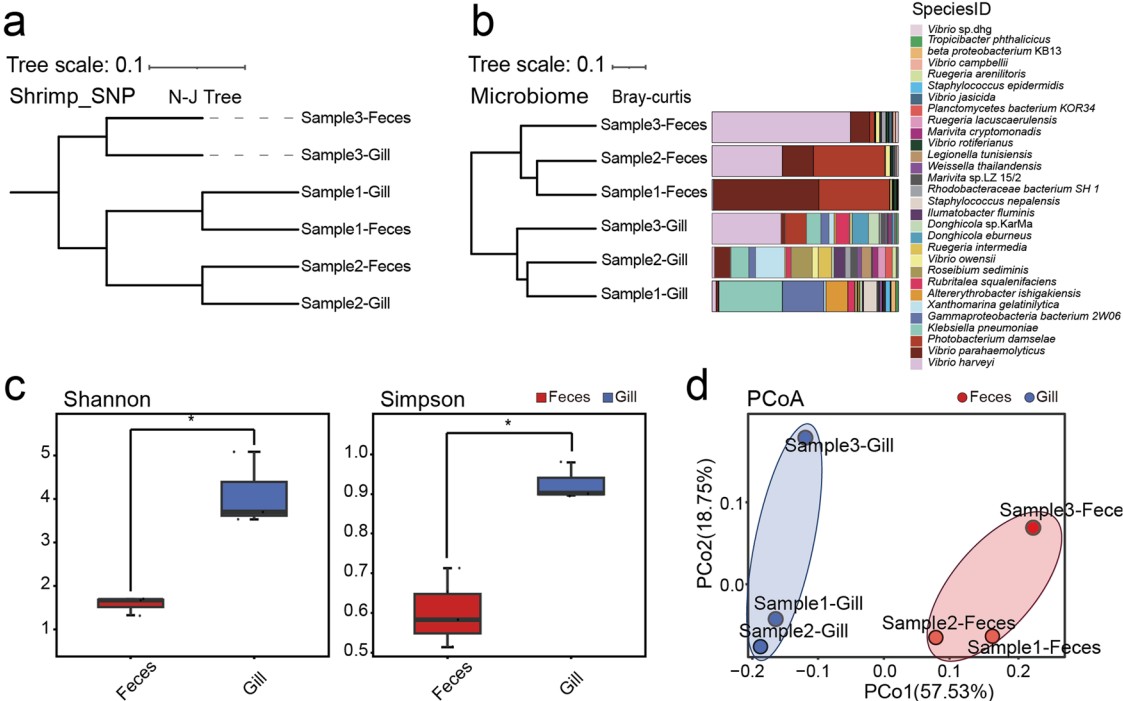

**Fig. 5 | Performance of hologenomic analysis in shrimp. a** Neighbor-joining clustering of gill and feces of three shrimps based on genetic distance. **b** Clustering analysis and profiling stack column of gill and feces of three samples at species level. **c** α-diversity of gill and feces (left: Shannon index; right: Simpson index). **d** Principal coordinate analysis (PCoA) of the microbial composition of gill and feces in three shrimps.

amount of lysozyme to the lysis solution based on our previous study to lyse the cytoderm of bacteria[47]. The swabs and filter paper that have been used for wiping gill of scallops were suspended in 100 μL lysis solution, containing 1×STE buffer (100 mM NaCl, 10 mM Tris-HCl, 1 mM EDTA, pH = 8.0), 1.0% SDS (Sigma, St Louis, MO, USA), 0.1 mg/mL protease K (Roche Diagnostics, Basel, Switzerland) and 0.05 mg lysozyme (TIAN-GEN, Beijing, China), then the solution was incubated at room temperature for 15 min. After the incubation, the solution was centrifuged at 12,000 rpm for 5 min, and the supernatant was transferred to a new centrifuge tube for the subsequent analysis.

**QIAamp® cador® Pathogen DNA process**. QIAamp® cador® Pathogen DNA process (Qiagen, Hidden, Germany) was used to extract DNA from CG sample and scallop larvae. For CG sample, different numbers of swabs (1/3/5/10) were tested with its efficacy of DNA extraction due to the fewer microbes obtained through wiping. The wiping samples were pretreated with 180 μL buffer ATL and 20 μL proteinase K, and after that incubated at 56 °C with constant agitation until the tissues were completely lysed. Then the lysate was transferred into the Pathogen Lysis Tube for 10 min vortex at maximum speed. After vortex, the lysate was transferred into a 2-ml microcentrifuge tube containing 20 μl proteinase K and 100 μl Buffer VXL and mixed by pulse vortex, then incubated at 20–25 °C for 15 min. The mix was transferred to the QIAamp Mini column for centrifuging at 6000×*g* (8000 rpm) for 1 min, then 600 μl Buffer AW1 was used to clean the membrane to remove organic matter. Final, 75 μl Buffer AVE was added to the QIAamp Mini column for centrifuging at full speed (20,000 xg; 14,000 rpm) for 1 min to gain the DNA. This method was also used to extract DNA from larvae samples.

**QIAamp® PowerFecal® DNA Kit**. DNA samples of feces were extracted by using QIAamp® PowerFecal® DNA Kit (Qiagen, Hidden, Germany). A total of 50 mg of feces/larvae were weighted and put into PowerBead Pro Tube, which containing 800 μL CD1, and then the mixture was mixed on Vortex Genie Mixer for 20 min. After mixing, DNA was extracted according to the steps described in the kit instructions. To

detect the integrity and yield of the DNA samples, all DNA samples were subjected to 1% agarose gel electrophoresis at 100 V for 25 min, and were viewed on a UV transilluminator instrument. Their yields were also measured by NanoDrop ND-1000 spectrophotometer (Thermo Fisher Scientific, Cleveland, OH, USA), in which DL was measured after purification

**Negative "kitome" analysis**. For negative control, the "kitome" experiments with two kits and sampling tools were conducted as follows: (1) cotton swabs combined with QIAamp® cador® Pathogen DNA process; (2) pipettes combined with QIAamp® PowerFecal® DNA Kit, by using the same protocol and under the same operating environment as in our experiments.

**Amplification of 16 s rRNA V4–5 region**
DNA samples were evaluated by DNA amplification of a short fragment of the V4–5 region of the 16 S rRNA. Each PCR was performed in a 20 μL reaction volume containing 0.5 μM of each primer (515 F:5'-GTGCCAG-CAGCCGCGGTAA, 907 R:5'-CCGTCAATTCCTTTGAGTTT), 50 ng of genomic DNA, 10 μL of Q5® High-Fidelity 2× Master Mix (NEB, Ipswich, MA). Cycling conditions consist of an initial denaturation step at 98 °C for 3 min, followed by 35 cycles of 98 °C for 20 s, 55 °C for 15 s and 72 °C for 20 s, and then a final extension step was performed at 72 °C for 5 min. The amplification products were subjected by agarose gel electrophoresis (1% agarose gel running at 100 V for 25 min) and viewed on a UV transilluminator instrument. All sequences were pre-processed following the standard QIIME (v.1.9) pipeline. Downstream bioinformatics analysis was performed using Parallel-Meta 3, a software package for comprehensive taxonomic and functional comparison of microbial communities. The clustering of OTUs was conducted at the 97% similarity level using a pre-clustered version of the Refseq database by BLASTN.

**Holo-2bRAD library construction**
DNA samples of gill, feces, and larvae were used to construct holo-2bRAD libraries following the protocol developed by our group[30,31]. In details,

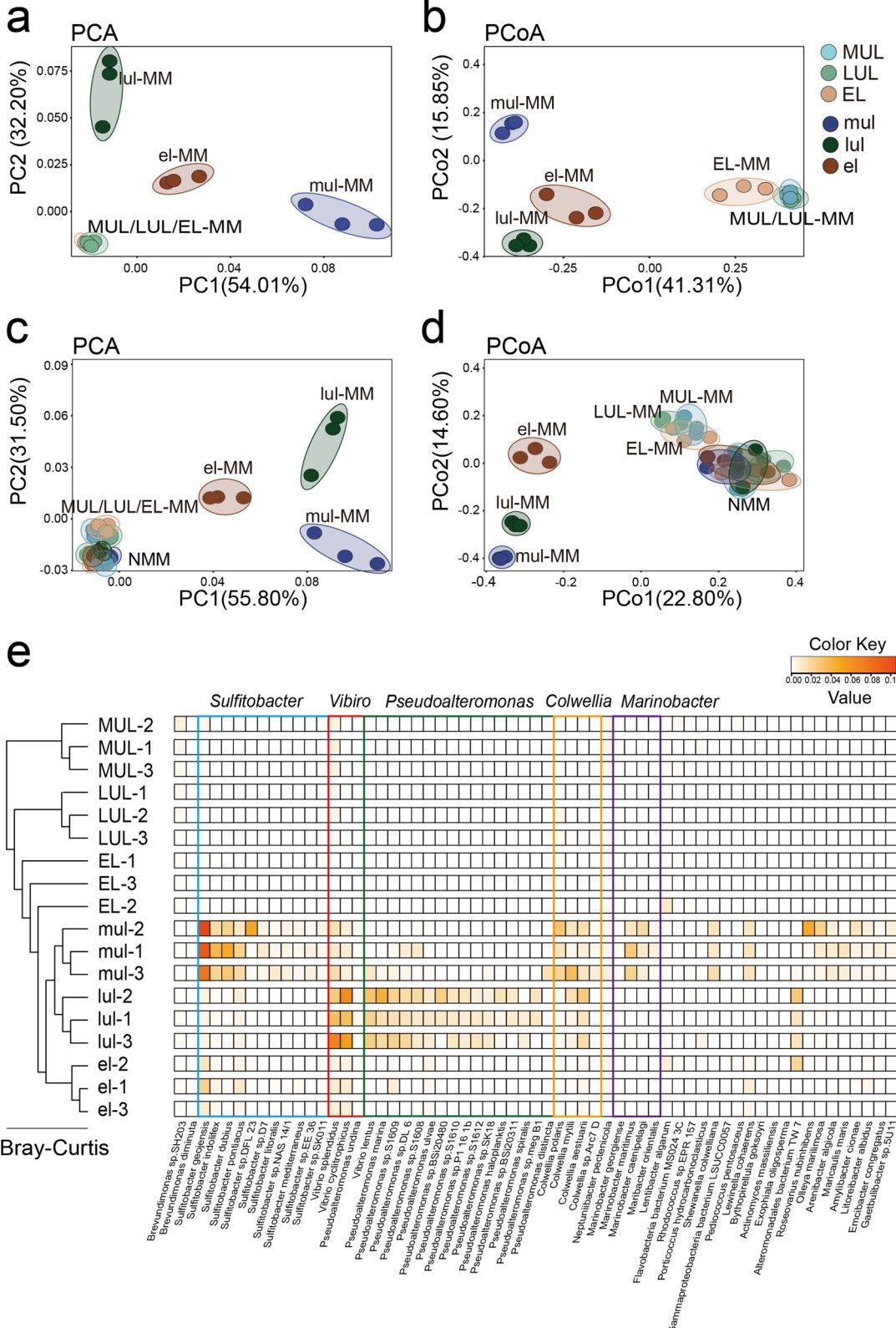

**Fig. 6 | Dynamic changes of microbiological compositions in scallop with larval viability.** β-diversity between different groups of 18 samples in MM (**a**, **b**), β-diversity between different groups of 36 samples in MM and NMM **c**, **d**, and the relative abundance of detected species in 18 samples of larvae in MM (**e**). MUL/mul, middle-umbo larvae; LUL/lul, late-umbo larvae, EL/el, eyespot larvae (uppercases represent larvae living in the upper layer, lowercases represent larvae living in the bottom layer. MM was the culturing hatchery which experienced mass mortalities of larvae, NMM was the culturing hatchery which did not).

~100 ng DNA was digested using 2 U BsaXI (NEB) in a total of 15 μL volume at 37 °C for 2 h, and then a total of 10 μL fragmented DNA were ligated with 1.8 μL of 5 μM adaptors using 200 U T4 DNA ligase (NEB) at 16 °C for 6 h in a 22 μL ligation solution. After ligation, PCR amplification was performed in a 60 μl reaction volume containing 1.6 μL of 10 μM primers, 0.6 mM dNTPs, 1× Phusion HF buffer and 0.8 U Phusion high-fidelity DNA polymerase (NEB) under the following PCR cycling conditions: an initial denaturation step at 98 °C for 5 min, then 16 cycles of 98 °C for 15 s, 60 °C for 10 s and 72 °C for 15 s, and a final extension step at 72 °C for 5 min. The PCR products were purified by 8% polyacrylamide gel and the gel-purified products were re-amplified for seven cycles using the same PCR program. The re-amplified products were purified by MinElute PCR purification kit and eluted using 15 μL of sterile deionized water to generate the final sequencing library. To check the reproducibility, two replicate libraries were constructed for each DNA sample. Meanwhile, the holo-2bRAD library constructed by DNA samples from the gill of the same individual of scallops and shrimps using a traditional phenol-chloroform method was set as control respectively. Quality control and concentration measurements were performed using Agilent 2100 Bioanalyzer DNA 1000 Chip and qPCR. The libraries were sequenced by using Illumina NovaSeq 6000 System (PE150 mode; Novagene Beijing, China).

### Data analysis

The raw sequencing reads were trimmed and quality filtered using the following steps: removing reads with ambiguous base calls (N), long homopolymer regions (>10 bp) or excessive low-quality bases (>20% of bases with quality score <10). The high-quality reads were then conducted to extract 2bRAD tags containing BsaXI recognition sites.

**Database construction**. We used the custom scripts to construct our hologenome database. Take the scallop as example, the predicted holo-2bRAD tags were initially extracted from the genome of *P. yessoensis*[64] and *L. Vannamei* (to be published) according to the recognition sites of the BsaXI, and de-redundancy was performed to obtain unique tags of scallop genome. Then, the microbial species-specific 2bRAD tags, which have no overlap with other species in 173,165 microbial genomes (downloaded from the NCBI RefSeq database) were download from GitHub (https://github.com/shihuang047/2bRAD-M). Finally, the predicted unique 2bRAD tags of scallop genome were combined with microbial species-specific tags, after then remove redundancy again to eliminate the interaction between redundant tags in the scallop genome and microbial genome. The database after secondary redundancy removal is the hologenome database of scallops, and the database construction process of other species is similar. The script used to build the database comes from 2bRAD-M pipeline[32] (https://github.com/shihuang047/2bRAD-M).

**Whole-genome genotyping of scallop using RADtyping program**. After preprocessing of the raw sequencing reads, RADtyping program: https://github.com/jinzhuangdou/RADtyping[39] was used to perform genome-wide genotyping of 2bRAD sequencing tags. The sequencing 2bRAD tags were aligned to the reference by using SOAP v.2.21[65], and genotyping of SNPs was conducted by codom_calling.pl pipeline implemented in the RADtyping package with default parameters. The detected sites were defined as the holo-2bRAD tag with a minimum depth of 1, and loci with the read coverage greater than 4× were considered as reliable genotypes, otherwise remarked as unknown genotypes. To identify specific loci in the scallop genome linked to variation in larvae-derived bacterial composition, a genome-wide association study (GWAS) was performed with GEMMA[66]. The SNP genotyping matrixes were explored to construct neighbor-joining (NJ) trees of individuals using PHYLIP v3.69[67] in shrimp and larvae of scallop. The NJ trees were visualized via iTOL v5[68] (https://itol.embl.de/).

**Analysis of microbial composition using 2bRAD-M pipeline**. 2bRAD-M pipeline[32] was used to conduct the analysis of microbial composition

(https://github.com/shihuang047/2bRAD-M). In brief, alignment of the sequencing holo-2bRAD tags to hologenome dataset and the analysis of taxonomic profiling and relative abundance were performed by using one custom script 2bRADM_Pipline.pl. The ggplot2, heatmap and venn implemented in R package (https://www.r-project.org/) were used to analysis accumulation of the relative abundance of different microorganisms within individuals, as well as species interaction and complement among different individuals or groups. The software of PAST[69] was used to estimate α- and β-diversity of the bacterial communities. The α-diversity metrics including Simpson and Shannon diversity index were calculated and visualized. The β-diversity analysis included principal component analysis (PCA) and principal coordinates analysis (PcoA), which was based on the Bray–Curtis metrics, were used to visualize the dissimilarity in bacterial community compositions of different samples. The relative abundance of different microorganisms (species) was used as a "response variable", the genotype of the host was used as "explanatory variable file". And different treatment groups were used as a "grouping file" for redundancy analysis (RDA), which was performed using Omicshare online tools (https://www.omicshare.com/tools/).

### Statistics and reproducibility

The statistical analyses conducted on the data and the sample sizes in each figure were described in their respective figure captions. Experiments were performed with at least two independent repeats. The original data can be found in the Supplementary Data file.

### Reporting summary

Further information on research design is available in the Nature Portfolio Reporting Summary linked to this article.

### Data availability

All the raw sequencing reads are deposited in the SRA (SUB13866949). The source data behind the graphs in the paper can be found in Supplementary Data.

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

## Acknowledgements
The authors acknowledge the grant support from National Key R&D Program of China (2022YFD2400301), National Natural Science Foundation of China (32130107 & 32002446), Marine S&T Fund of Shandong Province for Laoshan Laboratory (LSKJ202202803 & LSKJ202202804), Key R&D Project of Shandong Province (2021ZLGX03, 2022ZLGX01), and Taishan Scholar Project Fund of Shandong Province of China.

## Author contributions
S.W., P.L., Z.B., and J.H. designed the experiments and supervised the study. Q.X., Y.G., C.C., K.N., and X.D. performed the larvae sampling. C.M., C.X., T.Z., Q.M., Z.Y., and Z.X. carried out the experiments. C.M., C.X., P.L., Z.Y., J.L., and W.D. performed computational framework and analyzed the data. C.M. and P.L. wrote the manuscript. P.L. and S.W. revised the manuscript. The authors read and approved the final manuscript.

## Competing interests
The authors declare no competing interests.
