## [Peer Review File · Communications Biology]

Reviewers' comments:

Reviewer #1 (Remarks to the Author):

This is a delightfully rigorous paper, presenting a new very useful methodology described in great detail. I have no major concerns.

Minor things:

Check labels on Fig. 6 A: the group of higher-larvae is labeled NMM, aren't they all supposed to be MM?

L278 starts with "than" which I think needs to be deleted

Normalize the spelling throughout the paper: 2bRAD or 2b-RAD

I don't think anybody would ever trust a GWAS of just 18 samples (Fig 7 and associated text). This is also not directly related to the main theme of the paper (new method for microbiome composition analysis). I suggest removing it and replacing it with benchmarking the new method against the "standard" 16s amplicon analysis.

That said: Even though I would be pleased to see benchmarking against 16s, and I think this will make the paper even more convincing to the broad research community, I would be happy to see the paper published without it. Two reasons: one, I hate to request additional experiments when so much has been done already. And second, 16s is a "standard" method but it is more prone to biases than this one, so difference of 2bRAD results from 16s results, if detected, would most likely be due to biases in 16s approach.

Reviewer #2 (Remarks to the Author):

The study of Ma and colleagues proposes a new method for the joint assessment of microbiomes and their host genotype. The work is an important contribution and addresses a key technological limitation in the field. In addition, this study is an important step towards non-invasive animal research.

I believe that the manuscript would improve significantly by adding a description and discussion of the limitations of the holo-2bRAD approach. For example:

- The reliance on a database of complete (or near complete) genomes (for both host and microbes) could limit the ability to discover new taxa.
- How accurate can relative microbial abundances be estimated with the holo-2bRAD method? I

suspect this will be highly dependent on the number of cleavage sites, which could vary between species.

- Is there potential to look beyond taxonomic profiles? If not, what are the advantages of this method over 16S amplicon sequencing + traditional host genotyping?

Sections 2.2 and 2.3: the authors did a thorough analysis to identify the best sampling method. It should be noted that the DNA extraction kits were also different, which could explain the differences observed among sampling strategies. One key limitation is that there is no mention of negative controls. This is especially important in low-biomass samples. Could the results largely reflect the microbiomes associated with the kits and sampling tools used?

Section 2.4: Could you specify how many genomes (and from how many phyla) were used for the bacterial, archaeal and fungal kingdoms? The RefSeq database has a limited diversity of complete eukaryotic genomes, which would constrain the discovery of fungi.

Sections 2.7 and 2.8: These are very interesting analyses, but it was difficult to understand it at first. I would suggest starting with a short background, ideally supported by a figure, explaining: a) the life stages; b) the rationale for comparing upper/lower layers, and c) what is 'h1'?

Please also provide the scripts used to run these analyses. This is fundamental to ensure the reproducibility of the results.

Reviewer #3 (Remarks to the Author):

The hologenome is a concept of increasing importance in current research, making this manuscript timely and highly relevant. However, I have several comments regarding the aim and structure of the manuscript.

The study proposes a new approach to studying the hologenome using scallops and shrimps. However, it is unclear whether this is a methodological paper where scallops and shrimps are used as study models, or an article describing the hologenome of these organisms. The lack of clarity in organisation makes it challenging to follow the article, thus a more clarity in this sense it would be appreciated. For instance, the authors suggest that their study serves as a methodological paper, testing the proposed approach's utility in studying the hologenome (L91 and L413). They emphasise its potency as a tool, yet fail to compare it with other approaches, making difficult to assess the reliability of the approach.

Given the transition of gut microbiota research to shotgun sequencing, it is crucial to discuss the advantages and disadvantages of both 2bRAD and shotgun approaches and in which situation one approach should be selected over the other. I consider necessary to address the following points in the manuscript:

1) The authors claim that shotgun is high cost but it would be helpful if they could address the approximate cost difference between the two methods. Since 2bRDA requires complex data analysis (e.g., construction of hologenome database and generation of restriction fragments), it requires specialised bioinformatics tools and expertise, particularly for data processing, and downstream analysis. It appears to be very time-consuming and requires a highly skilled person to perform the analysis.

2) There should be discussion about the reliance of 2bRAD on specific recognition sites for fragmentation and the possible introduction of biases compared to shotgun sequencing.

3) The authors claim in this study that “Taxonomic profiling demonstrated that the microbiota of the gill contained a total of 11 phyla with 324 and 328 taxa in the two replicates”. However, since this manuscript lacks comparisons with other approaches, it is difficult to assess if this approach is capturing the entire microbial diversity.

4) How does the approach perform if the microbial community is unknown? In most of the wild species metagenomic studies, most of the Mags are lacking species-level annotation, confirming that they are new to science and they are not present in the database. How do you think this could affect the application of 2bRAD approach to other species? It is relevant to discuss this approach beyond scallops and shrimps.

5) The manuscript lacks statistical analyses to support claims of abundance changes in the microbial community. It is necessary to include more statistical analysis to test differences between groups.

6) It is essential to also emphasise that 2bRAD only allows for taxonomical characterisation, limiting functional insights crucial for understanding host-microbiota interactions.

7) Controls for bacterial contamination during sample collection and processing are not mentioned, a critical step for reliable results. Please add information about how the contamination was controlled in this study.

Therefore, considering all these points, I believe that the manuscript requires a complete reorganization. The authors should decide on the core ideas of their study and focus accordingly. For instance, if the manuscript aims to be methodological, more emphasis should be placed on demonstrating the method for hologenomics characterization and discussing its advantages and disadvantages compared to other methods. Conversely, if the focus is on the biology of scallops and shrimps, then information about the methodology should be appropriately placed within the methodology section and only mention the relevant methodological information in the main manuscript. I believe that this restructuring will enhance the clarity and coherence of the manuscript.

Below I provide comments that the authors might consider useful for a possible new version of the manuscript. The specific comments refer to line numbers as in the pdf.

L29: "for" should be deleted after "99.91%".

L58: A reference is needed after "this concept".

L60: I disagree with the statement that the "hologenome has dramatically increased in number." I believe the hologenome as a concept is gaining increasing relevance, but there are very few studies that analyze it. Therefore, I would suggest modifying the statement.

L86: It should include a short explanation of what 2b-RAD is.

L102: A reference is necessary at the end of the sentence.

L103: The paragraph about mass mortalities of bivalve larvae seems out of context.

L146: On what basis do you rely to assert that "the results still provide us with useful information"? How have you evaluated it?

L156: "Into" should be added after "enter".

L166: Please clarify why direct lysis was used for assessing differences between wiping materials and why other extraction methods were used for assessing the amount of swabs necessary.

L171-172: Confirm whether this test was conducted only in scallops.

L202: Replace "are" with "were".

L337: Provide complete statistical information.

L348-357: Statistical analyses testing microbiota differences at different taxonomic levels should be conducted.

L536: Write the species correctly.

Reviewer #1:

This is a delightfully rigorous paper, presenting a new very useful methodology described in great detail. I have no major concerns.

Response: We are very grateful for the reviewer's great interest and positive comments on our work.

Minor things:

Check labels on Fig. 6 A: the group of higher-larvae is labeled NMM, aren't they all supposed to be MM?

Response: We are very sorry for the error and thank you for pointing it out. We have corrected "NMM" to "MM" on Fig. 6A in the revised manuscript.

L278 starts with "than" which I think needs to be deleted

Response: Thanks for the suggestion, and we have deleted "than". Please see line 288 in the revised manuscript.

Normalize the spelling throughout the paper: 2bRAD or 2b-RAD

Response: Thanks for the suggestion, and we have standardized the description as holo-2bRAD in the revised manuscript.

I don't think anybody would ever trust a GWAS of just 18 samples (Fig 7 and associated text). This is also not directly related to the main theme of the paper (new method for microbiome composition analysis). I suggest removing it and replacing it with benchmarking the new method against the "standard" 16s amplicon analysis.

Response: Thanks for the valuable suggestion. Here, the GWAS analysis was mainly used to demonstrate that our approach has the power to conduct joint analysis of host and microbiota for hologenomic analysis. We agree that 18 individuals were a little weak for the GWAS analysis but still provide potential valuable preliminary results for follow-up validation studies. Therefore, we have moved it to supplement file in the revised manuscript (Fig. S5).

That said: Even though I would be pleased to see benchmarking against 16s, and I think this will make the paper even more convincing to the broad research community, I would be happy to see the paper published without it. Two reasons: one, I hate to request additional experiments when so much has been done already. And second, 16s is a "standard" method but it is more prone to biases than this one, so difference of 2bRAD results from 16s results, if detected, would most likely be due to biases in 16s

approach.

Response: Thanks for your valuable suggestion. We also appreciate your mentioning of the limitations of such comparison. In fact, 2bRAD approach and 16S amplicon sequencing have been systematically compared in our previous study (Sun *et al.*, 2022). Nevertheless, we also added the comparative analysis between 16S amplicon sequencing and holo-2bRAD, observing the high consistency (e.g. ~0.91) in the number and relative abundance of genus detected (Fig. S3). Please see section 2.5 in revised manuscript (line 297-302).

Fig. S3 The Venn diagram of genus detected and stack column of relative abundance between holo-2bRAD and 16S amplicon sequencing (A: scallop-feces; B: shrimp-feces; C: shrimp-gill).

Reference

Sun, Z. *et al.* Species-resolved sequencing of low-biomass or degraded microbiomes using 2bRAD-M. *Genome Biol.* **23**, 36 (2022).

Reviewer #2:

The study of Ma and colleagues proposes a new method for the joint assessment of microbiomes and their host genotype. The work is an important contribution and addresses a key technological limitation in the field. In addition, this study is an important step towards non-invasive animal research.

Response: We are very grateful for the reviewer's overall positive assessment of our work and thank his/her constructive comments for improving our manuscript. Please see below our response to the reviewer's specific comments.

I believe that the manuscript would improve significantly by adding a description

and discussion of the limitations of the holo-2bRAD approach. For example:

- *The reliance on a database of complete (or near complete) genomes (for both host and microbes) could limit the ability to discover new taxa.*

Response: Thanks for the valuable suggestion. The limitations of holo-2bRAD were added in the revised manuscript (line 333-337). The conduction of holo-2bRAD technology relies on existing databases, which limits its ability to discover new taxa, while this can be solved by assembling the microbial reference genomes of interested sample to cover the new taxa by using whole-metagenome shotgun (WMS) approach.

- *How accurate can relative microbial abundances be estimated with the holo-2bRAD method? I suspect this will be highly dependent on the number of cleavage sites, which could vary between species.*

Response: Thanks for the suggestion. For 25,260 species of bacteria in our database, averagely 2,199 cleavage sites were specific for each species. Despite the difference in the number of cleavage sites for each species, the accuracy of relative microbial abundances was also verified, including simulated data (see Fig. 2a in Sun *et al.*, 2022) and real samples (see Fig. 4a-f in Sun *et al.*, 2022). A simulated 50-species microbiome was generated and profiled by 2bRAD approach, observing the remarkably high precision (98.0%), recall (98.0%), and L2 similarity (96.9%) by comparing to the ground truth (see Fig. 2a in Sun *et al.*, 2022). To assess the performance of 2bRAD approach on real samples, 2bRAD approach, 16S rRNA sequencing, and shotgun sequencing on human fecal samples were performed and compared. Specifically, at the genus level, results of 2bRAD-M and 16S rRNA are highly consistent (mean Pearson correlation $R = 0.997$ and mean L2 similarity $L2 = 92.0\%$). As for the species level, we found that shotgun and 2bRAD sequencing data are also concordant in species-level profiling results, as evidenced by a high Pearson correlation ($R = 0.99$) and high L2 similarity (up to 97.8%).

Fig. 2a Simulated microbial community data consisting of 50 microbes were profiled by each of the 16 types IIB restriction enzymes. The scatter plots indicate the correlation of the taxonomic abundance estimated from 2bRAD-M with the expected abundance for each enzyme. The percentage number indicated in each plot represents the average genome coverage (compare to the original 50 microbial genomes) after digesting by the enzymes.

Fig. 4 a-c Comparison of taxonomic profiles at the genus level between 16S rRNA and 2bRAD-M. d-f Comparison of taxonomic profiles at the species level between 2bRAD-M and WMS. To

perform the comparison in a fair manner, we extracted 2bRAD reads from WMS data and then used these WMS originated 2bRAD reads as input for the 2bRAD-M computational pipeline. Then, species abundance generated by 2bRAD-M (using 2bRAD-M sequencing data) is used as the X-axis while the species abundance generated by 2bRAD-M (using WMS data) as the Y-axis for the scatter plot.

Reference

Sun, Z. *et al.* Species-resolved sequencing of low-biomass or degraded microbiomes using 2bRAD-M. *Genome biology*. 23, 36 (2022).

• *Is there potential to look beyond taxonomic profiles? If not, what is the advantages of this method over 16S amplicon sequencing + traditional host genotyping?*

Response: Yes. Holo-2bRAD obtains genetic information about hosts and microbiota through one sequencing library, thus it provides the great power to determine the relative ratio of microbial DNA to host DNA, which can track the fluctuations in the relative number of microorganisms in the host in different conditions, such as for animals before and after medication, sick and healthy individuals and the individuals in different stages (as mentioned in our manuscript section 2.6, line 345-347). Except that, compared with the resolution of 16S amplicon sequencing at the genus level and its ability to only identify bacteria, holo-2bRAD can accurately generate species-level bacterial, archaeal, and fungal profiles, even for these hard-to-sequence samples with low-biomass, high host DNA contamination, or severely fragmented DNA from degraded samples (Sun *et al.*, 2022).

Reference

Sun, Z. *et al.* Species-resolved sequencing of low-biomass or degraded microbiomes using 2bRAD-M. *Genome Biol.* 23, 36 (2022).

Sections 2.2 and 2.3: the authors did a thorough analysis to identify the best sampling method. It should be noted that the DNA extraction kits were also different, which could explain the differences observed among sampling strategies. One key limitation is that there is no mention of negative controls. This is especially important in low-biomass samples. Could the results largely reflect the microbiomes associated with the kits and sampling tools used?

Response: Thanks for the valuable suggestion. The "kitome" experiment was conducted as a negative control to avoid the impact of different types and batches of kits and sampling tools in the experiment. However, the concentration of "kitome" DNA was extremely low with 0.01~0.02 ng/ μ L, and under the same experimental conditions in our manuscript, the construction of sequencing library, including both the holo-2bRAD and 16S amplicon sequencing library, of "kitome" samples was not successful. This is why the sequencing results of "kitome" library did not appear in the manuscript. Since it was difficult to scale up microorganisms in "kitome" at the same number of cycles, the potential impact of "kitome" contamination in our samples can be neglected.

We have also added this in our revised manuscript, see line 263-269 and line 511-515.

Section 2.4: Could you specify how many genomes (and from how many phyla) were used for the bacterial, archaeal and fungal kingdoms? The RefSeq database has a limited diversity of complete eukaryotic genomes, which would constrain the discovery of fungi.

Response: In our manuscript, 173,165 microbial genomes from NCBI RefSeq (as of Oct 2019) were downloaded, representing 25,260 species of bacteria, 614 species of archaea and 289 species of eukaryote. After constructing the hologenome database using our approach, all species were retained in our database, indicating that no species were removed during the database construction. Our previous study has demonstrated the potential of 2b-RAD approach in discovery of fungi, with 5 phyla of fungi in our hologenome database. Up to now, 2b-RAD approach has been applied in the detection of fungi, including such as the survey in kindergarten microbiome (Lam *et al.*, 2022) and oral microbiota disorder in the GC patients (He *et al.*, 2023).

Reference

He S. *et al.* The oral microbiota disorder in the GC patients revealed by 2bRAD-M. *J. Transl. Med.* **21**. 831 (2023).

Lam, T. H. *et al.* Species-resolved metagenomics of kindergarten microbiomes reveal microbial admixture within sites and potential microbial hazards. *Front. Microbiol.* **13**, 871017 (2022).

Sections 2.7 and 2.8: These are very interesting analyses, but it was difficult to understand it at first. I would suggest starting with a short background, ideally supported by a figure, explaining: a) the life stages; b) the rationale for comparing upper/lower layers, and c) what is 'h1'?

Response: Thanks for your positive and constructive comments.

We have added a description and a figure (Fig. S4) of the life stages of the scallop larvae in the section 2.6 in the revised manuscript. The figure depicted (Fig. S4A) the development of umbo larvae (increase in shell diameter) and (Fig. S4B) the different lifestyles of larvae with different vitalities in the layer.

Bivalve larval mass mortality, a sudden massive loss of the bivalve stock (e.g., more than 30%; Soletchnik *et al.* 2007), occurred during the transition from the D-shaped to umbo larvae, often pose a threat to the marine fishery industry and ecosystems (Vaughn *et al.*, 2018). During this process, the sick larvae loss the swim ability and sunk to the bottom of the layer, while the healthy larvae suspended in the seawater. Therefore, we collected larvae with good vitality in the upper layer and sick larvae in the lower layer to explore the interaction mechanism between the host and microorganisms during the mass mortality of bivalve larvae.

We are apologized for the misspelled of 'h1', and has corrected to MM.

Fig. S4 Developmental processes in the umbo larvae of the scallop (A) and the different lifestyles of larvae with different vitalities in the layer (B). MUL/mul, middle umbo larvae; LUL/lul, late umbo larvae, EL/el, eyespot larvae (uppercases represent larvae living in the upper layer with good activity, lowercases represent larvae living in the bottom layer with poor activity).

References

- Soletchnik, P., Ropert, M., Mazurié, J., Fleury, P. G., & Le Coz, F. Relationships between oyster mortality patterns and environmental data from monitoring databases along the coasts of France. *Aquaculture* **271**, 384-400 (2007).
- Vaughn, C. C., & Hoellein, T. J. Bivalve impacts in freshwater and marine ecosystems. *Annu. Rev. Ecol. Evol. S.* **49**, 183-208 (2018).

Please also provide the scripts used to run these analyses. This is fundamental to ensure the reproducibility of the results.

Response: Thanks for the suggestion. The core analysis scripts of holo-2bRAD is the combination of 2bRAD-M pipeline and RADtyping program, and both were publicly available at Github: <https://github.com/shihuang047/2bRAD-M> and <https://github.com/jinzhuangdou/RADtyping>. Please see line 569 and 583 in the revised manuscript.

Reviewer #3 (Remarks to the Author):

The hologenome is a concept of increasing importance in current research, making this manuscript timely and highly relevant. However, I have several comments regarding the aim and structure of the manuscript.

Response: We appreciate the reviewer's positive comments of our manuscript and please see our detailed responses to his/her concerns below.

The study proposes a new approach to studying the hologenome using scallops and shrimps. However, it is unclear whether this is a methodological paper where scallops

and shrimps are used as study models, or an article describing the hologenome of these organisms. The lack of clarity in organisation makes it challenging to follow the article, thus a more clarity in this sense it would be appreciated. For instance, the authors suggest that their study serves as a methodological paper, testing the proposed approach's utility in studying the hologenome (L91 and L413). They emphasise its potency as a tool, yet fail to compare it with other approaches, making difficult to assess the reliability of the approach.

Response: Thanks for your valuable advice. Our manuscript serves as a methodological paper, and the validation of our method was carried out in scallops and shrimps. To make the theme of the manuscript clearer, we have reorganized the content of our manuscript. We have reduced the two parts describing the microbial species composition of scallops and shrimps to one part (please see part 2.5, line 253-337) and added the comparison of the holo-2bRAD with the 16S amplicon sequencing in the microbial analysis section as a validation of the reliability of our approach (line 298-302).

Given the transition of gut microbiota research to shotgun sequencing, it is crucial to discuss the advantages and disadvantages of both 2bRAD and shotgun approaches and in which situation one approach should be selected over the other. I consider necessary to address the following points in the manuscript:

1) The authors claim that shotgun is high cost but it would be helpful if they could address the approximate cost difference between the two methods. Since 2bRDA requires complex data analysis (e.g., construction of hologenome database and generation of restriction fragments), it requires specialised bioinformatics tools and expertise, particularly for data processing, and downstream analysis. It appears to be very time-consuming and requires a highly skilled person to perform the analysis.

Response: Thanks for your suggestion. In fact, the cost and bioinformatic efficiency comparison of both 2bRAD and shotgun approaches have been conducted in our previous study (Sun *et al.*, 2022). For the cost, only 1~5% of the sequencing data of shotgun approach were required by holo-2bRAD approach to produce a taxonomic profile of equivalent accuracy, resulting in a cost reduction of 20-100 folds (see Fig. 4g in Sun *et al.* 2022). The bioinformatic efficiency was evaluated in several aspects, including database storage and memory use, the runtime and operability. As for database storage and memory use, 2bRAD approaches require < 10-GB disk space to store the reference marker database and a relatively low RAM of 30 GB (equivalent to a desktop computer) as compared to Kraken2 and Bracken (see Fig. 2b in Sun *et al.* 2022). The runtime of 2b-RAD approaches, Kraken2 and MetaPhlAn4 was also compared using the same number of shotgun-sequencing reads in our test. 2bRAD approaches allowed species identification and calculation of the relative abundance of species within 0.5h, which is comparable to Kraken2 (~0.4h) while a quarter of MetaPhlAn4. Despite the requiring the construction of hologenome database, 2b-RAD

approaches take only one step to obtain the identified species and their relative abundance, comparable to the operability of commonly used software for shotgun sequencing (e.g. Kraken2). Moreover, 2bRAD approaches also make the improvement in algorithm with combining merits from both DNA-to-Marker and DNA-to-DNA methods to perform species identification and abundance estimation (Sun *et al.*,2023). The combined of two methods enable 2bRAD approaches to provide comprehensive genetic features such as species genome coverage, taxonomic count, and sequence count at the same time, laying a solid foundation for its excellent performance in eliminating false positives.

Fig. 4g The rarefaction analysis of 2bRAD-M and WMS samples. The species-level compositions in the subsampled data at each given sequencing depth were compared to the pre-rarefaction result for each method.

Fig. 2b Performance comparison of 2bRAD-M with Kraken2, Bracken, MetaPhlan2, and mOTUs2 based on 25 simulated communities. Two types of abundance are used as the ground truth of the simulation data to evaluate the performance: sequence abundance is used to evaluate Bracken and Kraken2, while taxonomic abundance is used to evaluate 2bRAD-M, mOTUs2, and MetaPhlan2.

References

Sun, Z. *et al.* Species-resolved sequencing of low-biomass or degraded microbiomes using 2bRAD-M. *Genome Biol.* **23**, 36 (2022).
 Sun, Z. *et al.* Removal of false positives in metagenomics-based taxonomy profiling via targeting

Type IIB restriction sites. *Nat. Commun.* **14**, 5321 (2023).

2) *There should be discussion about the reliance of 2bRAD on specific recognition sites for fragmentation and the possible introduction of biases compared to shotgun sequencing.*

Response: Thanks for the valuable suggestions. Holo-2bRAD indeed relied on the specific recognition sites for fragmentation, while the unbiased and broadly applicable representation of the microbial genomes by these IIB restriction cleavage sites has been verified in the previous study (see Fig. S2 in Sun *et al.*, 2022). Sequencing these IIB restriction cleavage sites does not bring obvious bias, which has been confirmed through a large amount of data, including simulated data (see Fig. 2a in Sun *et al.*, 2022) and real samples (see Fig. 4a-f in Sun *et al.*, 2022). A simulated 50-species microbiome was generated and profiled by 2bRAD approach, observing the remarkably high precision (98.0%), recall (98.0%), and L2 similarity (96.9%) by comparing to the ground truth (see Fig. 2a in Sun *et al.*, 2022). To assess the performance of 2bRAD approach on real samples, 2bRAD approach, 16S rRNA sequencing, and shotgun sequencing on human fecal samples were performed and compared. Specifically, at the genus level, results of 2bRAD-M and 16S rRNA are highly consistent (mean Pearson correlation $R = 0.997$ and mean L2 similarity $L2 = 92.0\%$). As for the species level, we found that shotgun and 2bRAD sequencing data are also concordant in species-level profiling results, as evidenced by a high Pearson correlation ($R = 0.99$) and high L2 similarity (up to 97.8%). Moreover, sequencing these IIB restriction cleavage sites, which were usually distributed far apart across a microbial genome significantly, mitigated the low-recall issue in the conventional DNA-to-Marker methods (Sun *et al.*, 2023). The last, as explained in earlier, 2bRAD approaches also make the improvement in algorithm with combining merits from both DNA-to-Marker and DNA-to-DNA methods to perform species identification and abundance estimation (Sun *et al.*, 2023), making it exhibit excellent performance in eliminating false positives.

Fig S2. The theoretical 2bRAD tags generated by 2bRAD-M and their originated genomes. Correlation of fragment size (left panel) or GC content (right panel) is shown. For a given genome, the collective size of all DNA tags cleaved by a type IIB restriction enzyme corresponds to a reduction in sequencing for one to two orders of magnitude (depending on genome size).

Fig. 2a Simulated microbial community data consisting of 50 microbes were profiled by each of the 16 types IIB restriction enzymes. The scatter plots indicate the correlation of the taxonomic abundance estimated from 2bRAD-M with the expected abundance for each enzyme. The percentage number indicated in each plot represents the average genome coverage (compare to the original 50 microbial genomes) after digesting by the enzymes.

Fig. 4 a–c Comparison of taxonomic profiles at the genus level between 16S rRNA and 2bRAD-M. d–f Comparison of taxonomic profiles at the species level between 2bRAD-M and WMS. To

perform the comparison in a fair manner, we extracted 2bRAD reads from WMS data and then used these WMS originated 2bRAD reads as input for the 2bRAD-M computational pipeline. Then, species abundance generated by 2bRAD-M (using 2bRAD-M sequencing data) is used as the X-axis while the species abundance generated by 2bRAD-M (using WMS data) as the Y-axis for the scatter plot.

References

Sun, Z. *et al.* Species-resolved sequencing of low-biomass or degraded microbiomes using 2bRAD-M. *Genome Biol.* **23**, 36 (2022).

Sun, Z. *et al.* Removal of false positives in metagenomics-based taxonomy profiling via targeting Type IIB restriction sites. *Nat. Commun.* **14**, 5321 (2023).

3) *The authors claim in this study that “Taxonomic profiling demonstrated that the microbiota of the gill contained a total of 11 phyla with 324 and 328 taxa in the two replicates”. However, since this manuscript lacks comparisons with other approaches, it is difficult to assess if this approaches is capturing the entire microbial diversity.*

Response: Thanks for your suggestion. As explained earlier, the comparison of 2bRAD and other approaches for microbiome analysis has been comprehensively validated in our previous study (Sun *et al.* 2022). In this study, we also added a comparison with 16S amplicon sequencing, and the high consistency of the genus detected (~90%) and the relative abundance (Pearson $r=0.9063-0.9367$) indicated that the holo-2bRAD was accurate for microbial analysis. The corresponding results have been added to the revised manuscript (please see line 298-302).

Reference

Sun, Z. *et al.* Species-resolved sequencing of low-biomass or degraded microbiomes using 2bRAD-M. *Genome Biol.* **23**, 36 (2022).

4) *How does the approach perform if the microbial community is unknown? In most of the wild species metagenomic studies, most of the Mags are lacking species-level annotation, confirming that they are new to science and they are not present in the database. How do you think this could affect the application of 2bRAD approach to other species? It is relevant to discuss this approach beyond scallops and shrimps.*

Response: Thanks for the suggestion. The conduction of holo-2bRAD technology relies on existing databases, which limits its ability to discover new taxa. To solve these limitations, it is usually recommended to use shotgun sequencing to de novo assemble the microbial genomes or MAGs of interested sample whose microbial community is unknown. The genomes or MAGs assembled by shotgun sequencing can be used for hologenome database construction, enabling holo-2bRAD to detect specific microbial species of this interested sample.

5) *The manuscript lacks statistical analyses to support claims of abundance changes in the microbial community. It is necessary to include more statistical analysis to test differences between groups.*

Response: Thanks for your suggestion. The statistical analysis was indeed necessary, and we have added these results in our revised manuscript, including t-test (line 312), Mann-Whitney U test (line 346-347) and Kruskal-Wallis test (see line 393-411).

6) It is essential to also emphasise that 2bRAD only allows for taxonomical characterisation, limiting functional insights crucial for understanding host-microbiota interactions.

Response: Thanks for the suggestion. Holo-2bRAD relies on the fragment of restriction enzyme for rather than the whole genome to perform microbial analysis, making it difficult directly to conduct the functional analysis. However, we can to some extent indirectly infer the function based on the whole genome corresponding to the fragment of restriction enzyme.

7) Controls for bacterial contamination during sample collection and processing are not mentioned, a critical step for reliable results. Please add information about how the contamination was controlled in this study.

Response: Thanks for your suggestion. First of all, all the samples were cultured and stabilized in sterile seawater (please see section 3.1) to control the potential contamination from seawater. Then, the "kitome" experiment was conducted as a negative control to avoid the impact of sampling tools and kits in the experiment. However, the concentration of "kitome" DNA was extremely low with 0.01~0.02 ng/ μ L, and under the same experimental conditions in our manuscript, the construction of sequencing library, including both the holo-2bRAD and 16S amplicon sequencing library, of "kitome" samples was not successful. This is why the sequencing results of "kitome" library did not appear in the manuscript. Since it was difficult to scale up microorganisms in "kitome" at the same number of cycles, we concluded the potential impact of "kitome" contamination in our samples can be neglected. We have also added this in our revised manuscript. Please see line 263-269 and line 511-515.

Therefore, considering all these points, I believe that the manuscript requires a complete reorganization. The authors should decide on the core ideas of their study and focus accordingly. For instance, if the manuscript aims to be methodological, more emphasis should be placed on demonstrating the method for hologenomics characterization and discussing its advantages and disadvantages compared to other methods. Conversely, if the focus is on the biology of scallops and shrimps, then information about the methodology should be appropriately placed within the methodology section and only mention the relevant methodological information in the main manuscript. I believe that this restructuring will enhance the clarity and coherence of the manuscript.

Response: Thanks for your constructive suggestions. We have made a substantial

reorganization of our manuscript, placing more emphasis to demonstrate our approach. We strengthen the methodological part with adding the comparison of holo-2bRAD with the other approaches and shortening the shrimp/gill part (see line 253-337 in our revised manuscript).

Below I provide comments that the authors might consider useful for a possible new version of the manuscript. The specific comments refer to line numbers as in the pdf.

L29: "for" should be deleted after "99.91%".

Response: Thank you for pointing it out. We have deleted "for" (see line 29).

L58: A reference is needed after "this concept".

Response: Thanks for your suggestion, and we have added the references (see line 57-58).

L60: I disagree with the statement that the "hologenome has dramatically increased in number." I believe the hologenome as a concept is gaining increasing relevance, but there are very few studies that analyze it. Therefore, I would suggest modifying the statement.

Response: Thanks for your suggestion, and we have replaced "dramatically" to "progressively" (see line 60).

L86: It should include a short explanation of what 2b-RAD is.

Response: We have added this explanation. Please see line 85-88 in the revised manuscript.

L102: A reference is necessary at the end of the sentence.

Response: We have added the references (please see line 103).

L103: The paragraph about mass mortalities of bivalve larvae seems out of context.

Response: Thanks for your suggestion, and we have added a background description of larvae mass mortalities. Please see line 103-106.

L146: On what basis do you rely to assert that "the results still provide us with useful information"? How have you evaluated it?

Response: Thanks for the suggestion. We apologize for the ambiguity in this description. Here, the result of this experiment has given some useful information on

the selection of the tissue enriched with microorganisms for hologenome analysis. We have added the detailed description to the revised manuscript (line 149-150).

L156: "Into" should be added after "enter".

Response: Thank you for pointing it out. We have added "into" after "enter" (see line 160).

L166: Please clarify why direct lysis was used for assessing differences between wiping materials and why other extraction methods were used for assessing the amount of swabs necessary.

Response: Thanks for the suggestion. The direct lysis method was used to evaluate the amount of biomass obtained from different wiping materials. However, the proportion of microorganisms in the samples obtained by wiping is extremely low, and it is necessary to use an optimized kit that enriches microbial DNA for hologenome analysis. In order to test the optimal biomass of this type of sample for the kit, the number of swabs was evaluated.

L171-172: Confirm whether this test was conducted only in scallops.

Response: Thanks for the suggestion. This test was conducted only in scallops. The evaluation results in scallop demonstrated that cotton swab was more suitable for wiping among the two wiping materials (e.g. cotton swab and filter paper), and the reason why the evaluation was not conducted in shrimp is that similar results have been reported in previous studies (Wortham *et al.*, 2020).

Reference

Wortham, J. L., & VanMaurik, L. Gill fouling in the economically important freshwater shrimp *Macrobrachium rosenbergii* (De) (Caridea: Palaemonidae). *J. Crustacean Biol.* **40**, 17-23 (2020).

L202: Replace "are" with "were".

Response: Thank you for pointing it out. We have corrected "are" to "were" (see line 208).

L337: Provide complete statistical information.

Response: Thanks for your valuable advice. The Mann-Whitney U test was performed to verify whether the difference between the two groups was significant. We have added this information in the revised manuscript (please see line 346-347).

L348-357: Statistical analyses testing microbiota differences at different taxonomic levels should be conducted.

Response: Thanks for your suggestion. We added statistical analyses to test microbiota differences at different taxonomic levels in the revised manuscript. Please see section 2.7 (line 393-411).

L536: Write the species correctly.

Response: Thank you for pointing it out. We have made the correction (see line 557).

REVIEWERS' COMMENTS:

Reviewer #2 (Remarks to the Author):

The authors have thoroughly revised the manuscript. In particular, it is now clear that negative controls were in place, the text now describes the advantages and potential limitations of the method clearly, and figure S4 is beautiful and informative. I am therefore supportive of this publication.

A few very minor suggestions to improve clarity:

257. Replace "similar raw reads" with "similar number of raw reads"

264: Rephrase to: (...) as a negative control to assess the potential impact of contamination (...)

266: Rephrase to: ... the construction of libraries was unsuccessful for both the holo-2bRAD and 16S amplicon sequencing.

317: 16S (with capital S)

335 - 337: This is a little confusing, and the paragraph above makes it clear that Holo-2bRAD focuses on taxonomy, so I think it is safe to remove this sentence.